# GLOBAL CONTEXT VISION TRANSFORMERS

## ABSTRACT

We propose global context vision transformer (GC ViT), a novel architecture that enhances parameter and compute utilization for computer vision tasks. The core of the novel model are global context self-attention modules, joint with standard local self-attention, to effectively yet efficiently model both long and short-range spatial interactions, as an alternative to complex operations such as an attention masks or local windows shifting. While the local self-attention modules are responsible for modeling short-range information, the global query tokens are shared across all global self-attention modules to interact with local key and values. In addition, we address the lack of inductive bias in ViTs and improve the modeling of inter-channel dependencies by proposing a novel downsampler which leverages a parameter-efficient fused inverted residual block. The proposed GC ViT achieves new state-of-the-art performance across image classification, object detection and semantic segmentation tasks. On ImageNet-1K dataset for classification, the tiny, small and base variants of GC ViT with 28M, 51M and 90M parameters achieve **83.4%**, **83.9%** and **84.4%** Top-1 accuracy, respectively, surpassing comparably-sized prior art such as CNN-based ConvNeXt and ViT-based Swin Transformer. Pre-trained GC ViT backbones in downstream tasks of object detection, instance segmentation, and semantic segmentation on MS COCO and ADE20K datasets outperform prior work consistently, sometimes by large margins.

## 1 INTRODUCTION

During the recent years, Transformers (Vaswani et al., 2017) have achieved State-Of-The-Art (SOTA) performance in Natural Language Processing (NLP) benchmarks and became the de facto model for various tasks. A key element in the success of Transformers is the self-attention mechanism which allows for capturing contextual representations via attending to both distant and nearby tokens (Yin et al., 2021). Following this trend, Vision Transformer (ViT) (Dosovitskiy et al., 2020) proposed to utilize image patches as tokens in a monolithic architecture with minor differences comparing to encoder of the original Transformer. Despite the historic dominance of Convolutional Neural Network (CNN) in computer vision, ViT-based models have achieved SOTA or competitive performance in various computer vision tasks. In essence, the self-attention mechanism in ViT allows for learning more uniform short and long-range information (Raghu et al., 2021) in comparison to CNN. However, the monolithic architecture of ViT and quadratic computational complexity of self-attention baffle their swift application to high resolution images (Yang et al., 2021a) in which capturing multi-scale long-range information is crucial for accurate representation modeling.

Several efforts (Liu et al., 2021; Dong et al., 2022; Chu et al., 2021a; Tu et al., 2022), most notably Swin Transformer (Liu et al., 2021), have attempted to address the balance between short- and long-range spatial dependencies by proposing multi-resolution architectures in which the self-attention is computed in local windows. In this paradigm, cross-window connections such as window shifting are used for modeling the interactions across different regions. Despite the progress, the limited receptive field of local windows challenges the capability of self-attention to capture long-range information, and window-connection schemes such as shifting only cover a small neighborhood in the vicinity of each window. Subsequent efforts such as Focal Transformer (Yang et al., 2021b) attempted to address this issue by designing highly sophisticated self-attention modules with increased model complexity.

In this work, we introduce the Global Context (GC) ViT network to address these limitations. Specifically, we propose a hierarchical ViT architecture consisting of local and global self-attention modules. At each stage, we compute global query tokens, using a novel fused inverted residual blocks,

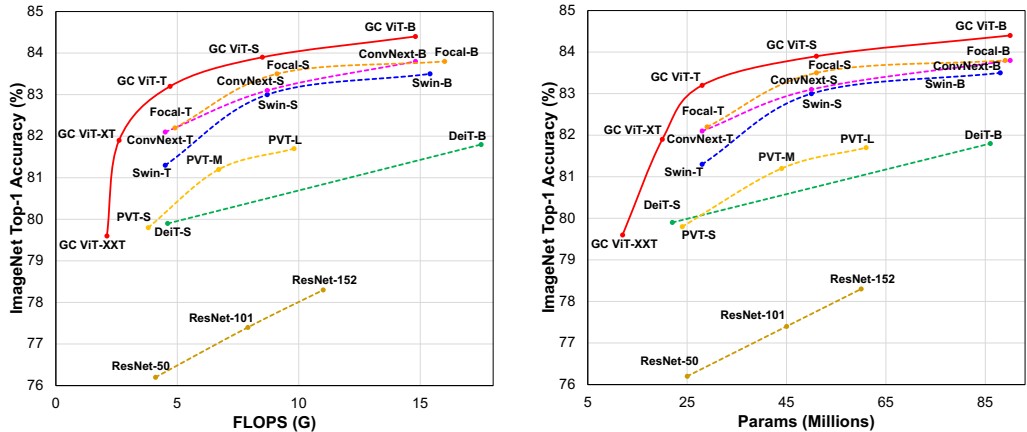

**Figure 1 – Top-1 accuracy *vs.* model FLOPs/parameter size on ImageNet-1K dataset.** GC ViT achieves new SOTA benchmarks for different model sizes as well as FLOPs, outperforming competing approaches by a significant margin. Best viewed in color.

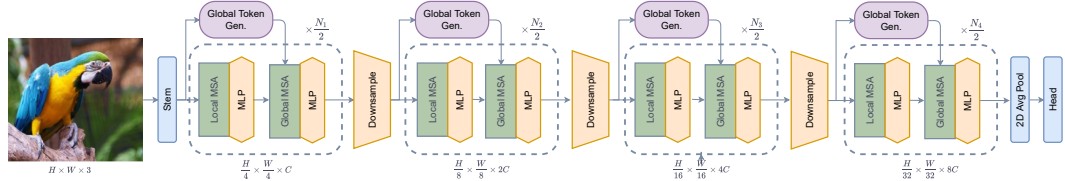

**Figure 2 –** Architecture of the proposed GC ViT. At each stage, a query generator extracts global query tokens which captures long-range information by interacting with local key and value representations. We use alternating blocks of local and global context self attention layers. Best viewed in color.

which we refer to as modified Fused-MBConv blocks, that encompass global contextual information from different image regions. While the local self-attention modules are responsible for modeling short-range information, the global query tokens are shared across all global self-attention modules to interact with local key and value representations. The design of our proposed framework for global query generator and self-attention is intuitive and simple and can be efficiently implemented using major deep learning framework. Hence, it eliminates sophisticated and computationally expensive operations and ensures the effectiveness of self-attention when applied to high-resolution images. In addition, we propose a novel downsampling block with a parameter-efficient fused-MBConv layer to address the lack of inductive bias in ViTs and enhancing the modeling of inter-channel dependencies.

We have extensively validated the effectiveness of the proposed GC ViT using three publicly available datasets for various computer vision tasks. For image classification using ImageNet-1K dataset, GC ViT with 28M, 51M, 90M and 201M parameters, referred to as tiny, small, base and large variants, achieve new SOTA benchmarks of **83.4%**, **83.9%**, **84.4%** and **84.6%** Top-1 accuracy. Hence, GC ViT consistently outperforms both ConvNeXt (Liu et al., 2022) and Swin Transformer (Liu et al., 2021) models by a significant margin (see Fig. 1). Using a pre-trained GC ViT base backbone with a Cascade Mask RCNN (He et al., 2017) head, our model achieves a box mAP of **52.9** for object detection and a mask mAP of **45.8** for instance segmentation on the MS COCO dataset. In addition, using an UPerNet (Xiao et al., 2018) head, our model achieves a mIoU of **49.0** on ADE20K for semantic segmentation. Other variants of GC ViT with different learning capacities also demonstrate SOTA results when compared to similarly-sized models on both MS COCO and ADE20K datasets. Hence, GC ViT demonstrates great scalability for high-resolution images on various downstream tasks, validating the effectiveness of the proposed framework in capturing both short and long-range information.

The main contributions of our work are summarized as follows:

- We introduce a compute and parameter-optimized hierarchical ViT with reparametrization of the design space (*e.g.*, embedding dimension, number of heads, MLP ratio).
- We design an efficient CNN-like token generator that encodes spatial features at different resolutions for global query representations.
- We propose global query tokens that can effectively capture contextual information in an efficient manner and model both local and global interactions.
- We introduce a parameter-efficient downsampling module with modified Fused MB-Conv blocks that not only integrates inductive bias but also enables the modeling of inter-channel dependencies.
- We demonstrate new SOTA benchmarks for : (1) ImageNet classification with Pareto fronts on ImageNet-1K for model size and FLOPs (see Fig. 1), and (2) downstream tasks such as detection, instance segmentation and semantic segmentation on MS COCO and ADE20K, respectively.

## 2 GC ViT ARCHITECTURE

**Architecture.** Fig. 2 depicts the architecture of GC ViT. We propose a hierarchical framework to obtain feature representations at several resolutions (called stages) by decreasing the spatial dimensions while expanding the embedding dimension, both by factors of 2. At first, given an input image with resolution of $\mathbf{x} \in \mathbb{R}^{H \times W \times 3}$, we obtain overlapping patches by applying a $3 \times 3$ convolutional layer with a stride of 2 and appropriate padding. Then patches are projected into a $C$-dimensional embedding space with another $3 \times 3$ convolutional layer with stride 2. Every GC ViT stage is composed of alternating local and global self-attention modules to extract spatial features. Both operate in local windows like Swin Transformer (Liu et al., 2021), however, the global self-attention has access to global features extracted by the global token generator. The token generator is a CNN-like module that extracts features from the entire image only once at every stage. After each stage, the spatial resolution is decreased by 2 while the number of channels is increased by 2 via a downsampling block. Resulting features are passed through average pooling and linear layers to create an embedding for a downstream task.

The GC ViT architecture benefits from novel blocks such as *a downsampling operator*, *a global query generator* and *a global self-attention module* described in the next sections.

**Downsampler.** We leverage an idea of spatial feature contraction from CNN models that imposes locality bias and cross channel interaction while reducing dimensions. We utilize a modified Fused-MBConv block, followed by a max pooling layer with a kernel size of 3 and stride of 2 as a downsampling operator, see Fig 3. The Fused-MBConv block in our work is similar to the one in EfficientNetV2 (Tan & Le, 2021) with modifications as in

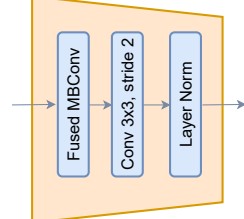

$$\begin{aligned}
\hat{\mathbf{x}} &= \text{DW-Conv}_{3\times3}(\mathbf{x}), \\
\hat{\mathbf{x}} &= \text{GELU}(\hat{\mathbf{x}}), \\
\hat{\mathbf{x}} &= \text{SE}(\hat{\mathbf{x}}), \\
\mathbf{x} &= \text{Conv}_{1\times1}(\hat{\mathbf{x}}) + \mathbf{x},
\end{aligned} \tag{1}$$

**Figure 3** – Downsampling block for dimension reduction.

where SE, GELU and DW-Conv$_{3\times3}$ denote Squeeze and Excitation block (Hu et al., 2018), Gaussian Error Linear Unit (Hendrycks & Gimpel, 2016) and $3 \times 3$ depth-wise convolution, respectively. In our proposed architecture, the Fused-MBConv blocks provide desirable properties such as inductive bias and modeling of inter-channel dependencies. It is ablated in Table 5.

**Attention.** Multi-head self-attention is the the core computational operator in the GC ViT architecture, it extracts semantic information from the image. GC ViT is composed of local and global self-attention modules illustrated in Fig 4. Similar to Swin Transformer (Liu et al., 2021), we benefit from splitting the image into windows and performing local self-attention within them, this leads to linear complexity scaling with image size. The local self-attention extracts local, short-range, information and in order to facilitate long range dependencies we propose to use a novel global self attention to allow cross-patch communication with those far beyond the local window. Global self-attention attends other regions in the image via global query token that represents image embedding extracted with CNN-like module.

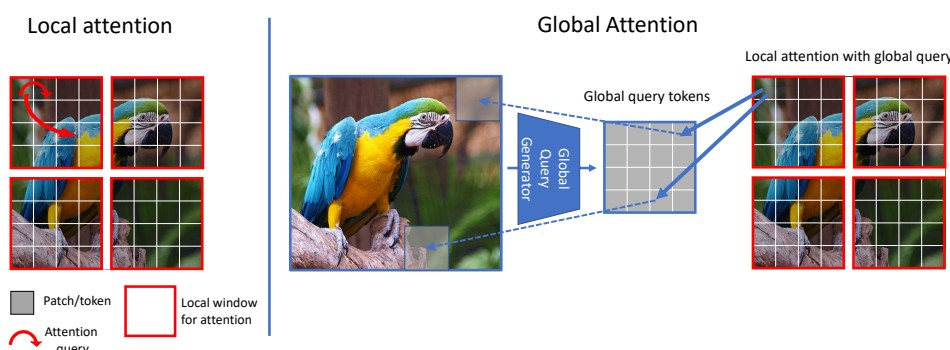

**Figure 4** – Attention formulation. Local attention is computed on feature patches within local window only (left). On the other hand, the global features are extracted from the entire input features and then repeated to form global query tokens. The global query is interacted with local key and value tokens, hence allowing to capture long-range information via cross-region interaction. Best viewed in color.

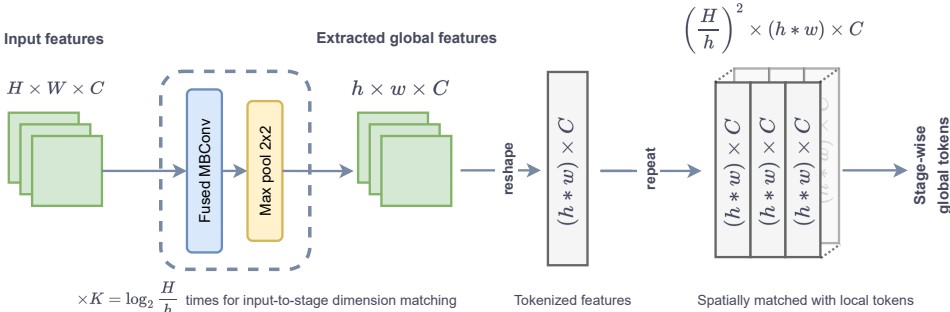

**Figure 5** – Global query generator schematic diagram. It is designed to (i) transform an input feature map to the current stage of dimension $H, W, C$ denoting height, width, and channel respectively, (ii) extract features via repeating the modified Fused MBConv block, joint with down-sampling, $\log_2 \frac{H}{h}$ times for dimension matching to local window size $h$ (iii) output is reshaped and repeated to $\left(\frac{H}{h}\right)^2$ number of local tokens that can attend to global contextual information. $\star$ denotes merged dimensions during reshaping.

## 2.1 GLOBAL QUERY GENERATOR

We propose to generate global query tokens that encompass information across the entire input feature maps for interaction with local key and value feature pairs. Specifically, as shown in Fig. 5, a layer **f** in the the generator consists of a Fused-MBConv block followed by a max pooling layer, similar to the one described in Sec. 2, and the final global query $\mathbf{q_{g,i}}$ at stage $i$ ($i \in \{1, 2, 3, 4\}$) of GC ViT is computed according to

$$
\begin{aligned}
\mathbf{x}^i &= \text{F-MBConv}(\mathbf{x}^{i-1}), \\
\mathbf{x}^i &= \text{MaxPool}(\mathbf{x}^i).
\end{aligned}
\tag{2}
$$

These query tokens are computed once at every stage of the model and shared across all global attention blocks, hence decreasing number of parameters and FLOPs and improving the generalizability. In addition, the global attention layers only learn local key and value features which will be used for interaction with global query tokens.

## 2.2 GLOBAL SELF-ATTENTION

Fig. 4 demonstrates the main idea behind our contribution. Local self-attention can only query patches within a local window, whereas the global attention can query different image regions while still operating within the window. At each stage, the global query component is pre-computed as described in Sec.2.1. The global self-attention utilizes the extracted global query tokens, obtained according to Eq. 2 and shared across all blocks, to interact with the local key and value representations. In addition, GC ViT employs alternating local and global self-attention blocks to effectively capture

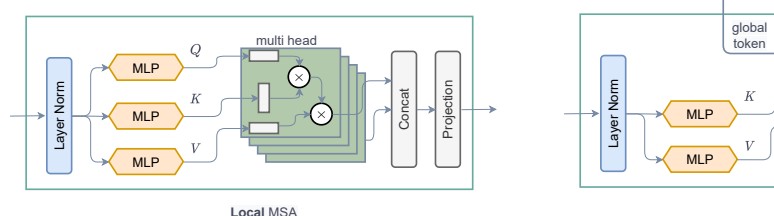

**Figure 6** – Local and global attention blocks. Global attention block does not compute query vector and reuses global query computed via Global Token Generation.

both local and global spatial information. Fig. 6 illustrates the difference between local and global self-attention. The global attention query $\mathbf{q_g}$ has a size of $B \times C \times h \times w$, wherein $B$, $C$, $h$ and $w$ denote batch size, embedding dimension, local window height and width, respectively. Moreover, $\mathbf{q_g}$ is repeated along the batch dimension to compensate for the overall number of windows and batch size $B^* = B \times N$ where $N$ is the number of local windows. $\mathbf{q_g}$ is further reshaped into multiple heads. The value and key are computed within each local window using a linear layer. The global self-attention query, key and value features are computed as follows

$$\mathbf{Q_g} \in \mathbb{R}^{B^* \times C \times h \times w} := [\mathbf{q_g}, ..., \mathbf{q_g}], \mathbf{q_g} \in \mathbb{R}^{B \times C \times h \times w}, \tag{3}$$

$$\mathbf{q_g} \in \mathbb{R}^{B^* \times N \times C} \xleftarrow{\text{reshape}} \mathbf{Q_g} \in \mathbb{R}^{B^* \times C \times h \times w}, \tag{4}$$

$$\mathbf{k}, \mathbf{v} = \mathbf{g}(\mathbf{x}) \in \mathbb{R}^{B^* \times N \times C}. \tag{5}$$

Since the partitioned windows only contain local information, interaction with rich contextual information embedded in the global query tokens provides an effective way of enlarging the receptive field and attending to various regions in the input feature maps. The self-attention module is computed as in

$$\text{Attention}(\mathbf{q_g}, \mathbf{k}, \mathbf{v}) = \text{Softmax}(\frac{\mathbf{q_g} \mathbf{k}}{\sqrt{\mathbf{d}}} + \mathbf{b})\mathbf{v}, \tag{6}$$

where $\mathbf{d}$ is scaling factor and $\mathbf{b}$ is a learnable relative position bias term. Assuming position change between $[-p+1, p-1]$ along horizontal and vertical axes, $\mathbf{b}$ is sampled from the grid $\hat{\mathbf{b}} \in \mathbb{R}^{(2p-1) \times (2p-1)}$. As shown in Sec. 5, relative position bias improves the performance, especially for dense prediction downstream tasks. In Algorithm 1, we present a PyTorch-like pseudocode for computing global self-attention in GC ViT. A complexity analysis of the global self-attention is presented in the supplementary materials.

**Algorithm. 1** Global Attention Pseudocode in PyTorch Style

```
# Input/output shape: (B*, N, C)
# B*: Batchsize*Num Windows; H:
    Height;
# W: Width; C: dim; q_g: Global Token
    ;
# F: Num Attention Head; N: Num
    Windows;
def init():
    f = nn.Linear(C, 2*C)
    softmax = nn.Softmax(dim=-1)

def forward(x, q_g):
    B*, N, C = x.shape
    B, C, h, w = q_g.shape
    kv = f(x).reshape(B*, N, 2, F, C
        // F)
    kv = kv.permute(2, 0, 3, 1, 4)
    k, v = split(kv, (1, 1), 0)
    q_g = q_g.repeat(1, B* // B, 1, 1)
    q_g = q_g.reshape(B*, F, N, C // F
        )
    qk = matmul(q_g, k.transpose(-2,
        -1))
    attn = softmax(qk)
    return matmul(attn, v).reshape(B*,
        N, C)
```

## 3  RELATED WORK

**ViT**. The ViT (Dosovitskiy et al., 2020) was first proposed as an alternative to CNNs with the advantage of enlarged receptive field, due to its self-attention layers. However, it lacked desirable properties of CNNs such as inductive biases and translation invariance and required large-scale training datasets to achieve competitive performance. Data-efficient Image Transformers (DeiT) Touvron et al. (2021b) introduced a distillation-based training strategy which significantly improved the classification accuracy. LeViT (Graham et al., 2021) proposed a hybrid model with re-designed multi-layer perceptron (MLP) and self-attention modules that are highly-optimized for fast inference. Cross-covariance Image Transformer (XCiT) (Ali et al., 2021) introduced a transposed self-attention module for

modeling the interactions of feature channels. Convolutional vision Transformer (CvT) (Wu et al., 2021) introduced convolutional token embedding layer and Transformer block in a hierarchical architecture to improve the efficiency and accuracy of ViTs. Conditional Position encoding Vision Transformer (CPVT) (Chu et al., 2021b) demonstrated improved performance on various tasks such as image classification and object detection by conditioning the position encoding on localized patch token. Tokens-To-Token Vision Transformer (T2T-ViT) (Yuan et al., 2021) proposed a transformation layer for aggregating adjacent tokens and establishing image prior by exploiting spatial correlations. Pyramid Vision Transformer (PVT) (Wang et al., 2021) proposed a hierarchical architecture with patch embedding at the beginning of each stage and spatial dimension reduction to improve the computational efficiency. Independently, Swin Transformers (Liu et al., 2021) also proposed a hierarchical architecture in which self-attention is computed within local windows which are shifted for region interaction. Twins Transformer (Chu et al., 2021a) proposed a spatially separable self-attention with locally-grouped and global sub-sampling modules to improve the efficiency. Focal Transformer (Yang et al., 2021b) introduced the Focal self-attention to capture long-range spatial interactions. PVT-v2 (Wang et al., 2022) improved performance and efficiency comparing to PVT (Wang et al., 2021) by introducing overlapping patch embedding, convolutional feed-forward network and linear attention. EdgeViT (Pan et al., 2022) introduced a lightweight ViT model, with global sparse attention and local aggregations and propagation modules for capturing short and long-range information

**ConvNet**. Since the advent of deep learning, CNNs (Krizhevsky et al., 2012; Simonyan & Zisserman, 2014; Howard et al., 2017; He et al., 2016; Szegedy et al., 2016; Huang et al., 2017; Hu et al., 2018) have dominated computer vision benchmarks with SOTA performance. Recently, inspired by ViTs, ConvMixer (Trockman & Kolter, 2022) introduced a simple architecture with large-kernel depth-wise and point-wise convolutional layers and global pooling with competitive performance for classification. Furthermore, ConvNeXt (Liu et al., 2022) proposed modifications to the architecture of ResNet (He et al., 2016), and achieved competitive benchmarks for classification, detection and segmentation tasks.

## 4 EXPERIMENTS

**Image classification.** For image classification, we trained and tested our model on ImageNet-1K dataset (Deng et al., 2009). To allow for a fair comparison, all GC ViT variants are trained by following training configurations of previous efforts (Liu et al., 2021; Yang et al., 2021b; Chu et al., 2021a). Specifically, all models are trained on 4 nodes (32 A100 GPUs) with the AdamW (Loshchilov & Hutter, 2017) optimizer for 300 epochs with an initial learning rate of 0.001, weight decay of 0.05, cosine decay scheduler and 20 warm-up and cool-down epochs, respectively. We use total batch sizes of 4096 for GC ViT-XXT, GC ViT-XT and GC ViT-T models and 1028 for all other variants. See supplementary materials for more training details.

**Object detection and semantic segmentation** For object detection and instance segmentation, we trained our model on MS COCO (Lin et al., 2014) with a Mask-RCNN (He et al., 2017) head, using ×3 LR schedule with an initial learning rate of 0.0001, a batch size of 16 and weight decay of 0.05. Following Liu et al. (2022), we compared against Tiny, Small and Base model variants using Cascade Mask-RCNN but only compared against Tiny variant using Mask-RCNN. For semantic segmentation, we used the ADE20K dataset (Zhou et al., 2017) with a UPerNet (Xiao et al., 2018) segmentation head. Following previous efforts, we used a random crop size of $512 \times 512$ for the input images. For fair assessment, we only compare against models with a pre-trained ImageNet-1K backbone.

### 4.1 CLASSIFICATION

We present the ImageNet-1K classification benchmarks in Table 1 and compare against CNN, ViT-based models across different model sizes. Our model achieves new SOTA benchmarks in all categories. Specifically, the proposed GC ViT surpasses similar-sized counterpart models by +0.5% for GC ViT-XT (82.0%) compared to T2T-ViT-14 (Yuan et al., 2021) (81.5%), +0.7% for GC ViT-T (83.4%) over CSwin-T (Dong et al., 2022) (82.7%), +0.3% for GC ViT-S (83.9%) over CSwin-S (Dong et al., 2022) (83.6%), +0.2% for GC ViT-B (84.4%) compared to CSwin-B (Dong et al., 2022) (84.2%) and +0.1% for GC ViT-L (84.6%) over CoAtNet-3 (Dai et al., 2021) (84.5%), respectively. Furthermore, as shown in Fig. 1, GC ViT models have better or comparable computational efficiency in terms of number FLOPs over the competing counterpart models.

**Table 1** – Image classification benchmarks on **ImageNet-1K** dataset (Deng et al., 2009).

| Method | Param (M) | FLOPs (G) | Image Size | Top-1 (%) |
|---|---|---|---|---|
| ResMLP-S12 (Touvron et al., 2021a) | 15 | 3.0 | $224^2$ | 76.6 |
| PVT-v2-B1 (Wang et al., 2022) | 13 | 2.1 | $224^2$ | 78.7 |
| **GC ViT-XXT** | 12 | 2.1 | $224^2$ | 79.8 |
| EdgeViT-S (Pan et al., 2022) | 11 | 1.9 | $224^2$ | **81.0** |
| DeiT-Small/16 (Touvron et al., 2021b) | 22 | 4.6 | $224^2$ | 79.9 |
| T2T-ViT-14 (Yuan et al., 2021) | 22 | 5.2 | $224^2$ | 81.5 |
| **GC ViT-XT** | 20 | 2.6 | $224^2$ | **82.0** |
| ResNet50 (He et al., 2016) | 25 | 4.1 | $224^2$ | 76.1 |
| Swin-T (Liu et al., 2021) | 29 | 4.5 | $224^2$ | 81.3 |
| CoAtNet-0 (Dai et al., 2021) | 25 | 4.2 | $224^2$ | 81.6 |
| PVT-v2-B2 (Wang et al., 2022) | 25 | 4.0 | $224^2$ | 82.0 |
| ConvNeXt-T (Liu et al., 2022) | 29 | 4.5 | $224^2$ | 82.1 |
| Focal-T (Yang et al., 2021b) | 29 | 4.9 | $224^2$ | 82.2 |
| CSwin-T (Dong et al., 2022) | 23 | 4.3 | $224^2$ | 82.7 |
| **GC ViT-T** | 28 | 4.7 | $224^2$ | **83.4** |
| ResNet-101 (He et al., 2016) | 44 | 7.9 | $224^2$ | 77.4 |
| Swin-S (Liu et al., 2021) | 50 | 8.7 | $224^2$ | 83.0 |
| ConvNeXt-S (Liu et al., 2022) | 50 | 8.7 | $224^2$ | 83.1 |
| CoAtNet-1 (Dai et al., 2021) | 42 | 8.4 | $224^2$ | 83.3 |
| Focal-S (Yang et al., 2021b) | 51 | 9.1 | $224^2$ | 83.5 |
| CSwin-S (Dong et al., 2022) | 35 | 6.9 | $224^2$ | 83.6 |
| **GC ViT-S** | 51 | 8.5 | $224^2$ | **83.9** |
| ResNet-152 (He et al., 2016) | 60 | 11.6 | $224^2$ | 78.3 |
| ViT-Base/16 (Dosovitskiy et al., 2020) | 86 | 17.6 | $224^2$ | 77.9 |
| DeiT-Base/16 (Touvron et al., 2021b) | 86 | 17.6 | $224^2$ | 81.8 |
| Swin-B (Liu et al., 2021) | 88 | 15.4 | $224^2$ | 83.3 |
| CoAtNet-2 (Dai et al., 2021) | 42 | 8.4 | $224^2$ | 83.3 |
| ConvNeXt-B (Liu et al., 2022) | 89 | 15.4 | $224^2$ | 83.8 |
| Focal-B (Yang et al., 2021b) | 90 | 16.0 | $224^2$ | 83.8 |
| PVT-v2-B5 (Wang et al., 2022) | 82 | 11.8 | $224^2$ | 83.8 |
| CSwin-B (Dong et al., 2022) | 78 | 15.0 | $224^2$ | 84.2 |
| BoTNet (Dong et al., 2022) | 79 | 19.3 | $256^2$ | 84.2 |
| **GC ViT-B** | 90 | 14.8 | $224^2$ | **84.4** |
| ConvNeXt-L (Liu et al., 2022) | 198 | 34.4 | $224^2$ | 84.3 |
| CoAtNet-3 (Dai et al., 2021) | 168 | 34.7 | $224^2$ | 84.5 |
| **GC ViT-L** | 201 | 32.6 | $224^2$ | **84.6** |

**Table 2** – Object detection and instance segmentation benchmarks using Mask R-CNN and Cascade Mask R-CNN on **MS COCO** dataset (Lin et al., 2014). All models employ $3\times$ schedule.

| Backbone | Param (M) | FLOPs (G) | $AP^{box}$ | $AP^{box}_{50}$ | $AP^{box}_{75}$ | $AP^{mask}$ | $AP^{mask}_{50}$ | $AP^{mask}_{75}$ |
|---|---|---|---|---|---|---|---|---|
| Mask-RCNN $3\times$ schedule | | | | | | | | |
| Swin-T (Liu et al., 2021) | 48 | 267 | 46.0 | 68.1 | 50.3 | 41.6 | 65.1 | 44.9 |
| ConvNeXt-T (Liu et al., 2022) | 48 | 262 | 46.2 | 67.9 | 50.8 | 41.7 | 65.0 | 44.9 |
| **GC ViT-T** | 48 | 291 | **47.9** | **70.1** | **52.8** | **43.2** | **67.0** | **46.7** |
| Cascade Mask-RCNN $3\times$ schedule | | | | | | | | |
| DeiT-Small/16 (Touvron et al., 2021b) | 80 | 889 | 48.0 | 67.2 | 51.7 | 41.4 | 64.2 | 44.3 |
| ResNet-50 (He et al., 2016) | 82 | 739 | 46.3 | 64.3 | 50.5 | 40.1 | 61.7 | 43.4 |
| Swin-T (Liu et al., 2021) | 86 | 745 | 50.4 | 69.2 | 54.7 | 43.7 | 66.6 | 47.3 |
| ConvNeXt-T (Liu et al., 2022) | 86 | 741 | 50.4 | 69.1 | 54.8 | 43.7 | 66.5 | 47.3 |
| **GC ViT-T** | 85 | 770 | **51.6** | **70.4** | **56.1** | **44.6** | **67.8** | **48.3** |
| X101-32 (Xie et al., 2017) | 101 | 819 | 48.1 | 66.5 | 52.4 | 41.6 | 63.9 | 45.2 |
| Swin-S (Liu et al., 2021) | 107 | 838 | 51.9 | 70.7 | 56.3 | 45.0 | 68.2 | 48.8 |
| ConvNeXt-S (Liu et al., 2022) | 108 | 827 | 51.9 | 70.8 | 56.5 | 45.0 | 68.4 | 49.1 |
| **GC ViT-S** | 108 | 866 | **52.4** | **71.0** | **57.1** | **45.4** | **68.5** | **49.3** |
| X101-64 (Xie et al., 2017) | 140 | 972 | 48.3 | 66.4 | 52.3 | 41.7 | 64.0 | 45.1 |
| Swin-B (Liu et al., 2021) | 145 | 982 | 51.9 | 70.5 | 56.4 | 45.0 | 68.1 | 48.9 |
| ConvNeXt-B (Liu et al., 2022) | 146 | 964 | 52.7 | 71.3 | 57.2 | 45.6 | 68.9 | 49.5 |
| **GC ViT-B** | 146 | 1018 | **52.9** | **71.7** | **57.8** | **45.8** | **69.2** | **49.8** |

## 4.2 DETECTION

In Table 2, we present object detection and instance segmentation benchmarks on MS COCO dataset. Using a Mask-RCNN head, the model with pre-trained GC ViT-T (47.9/43.2) backbone outperforms counterparts with pre-trained ConvNeXt-T (Liu et al., 2022) (46.2/41.7) by +1.7 and +1.5 and Swin-T (Liu et al., 2021) (46.0/41.6) by +1.9 and +1.6 in terms of box AP and mask AP, respectively. Using a Cascade Mask-RCNN head, the models with pre-trained GC ViT-T (51.6/44.6) and GC ViT-S (52.4/45.4) backbones outperform ConvNeXt-T (Liu et al., 2022) (50.4/43.7) by +1.2 and +0.9 and ConvNeXt-S (Liu et al., 2022) (51.9/45.0) by +0.5 and +0.4 in terms of box AP and mask AP, respectively. Furthermore, the model with

| Backbone | Param (M) | FLOPs (G) | mIoU |
|---|---|---|---|
| DeiT-Small/16 (Touvron et al., 2021b) | 52 | 1099 | 44.0 |
| Swin-T (Liu et al., 2021) | 60 | 945 | 44.5 |
| ResNet-101 (He et al., 2016) | 86 | 1029 | 44.9 |
| Twins-SVT-S (Chu et al., 2021a) | 55 | - | 46.2 |
| **GC ViT-T** | 58 | 947 | **47.0** |
| Swin-S (Liu et al., 2021) | 81 | 1038 | 47.6 |
| Twins-SVT-B (Chu et al., 2021a) | 89 | - | 47.7 |
| **GC ViT-S** | 84 | 1163 | **48.3** |
| Swin-B (Liu et al., 2021) | 121 | 1188 | 48.1 |
| Twins-SVT-L (Chu et al., 2021a) | 133 | - | 48.8 |
| **GC ViT-B** | 125 | 1348 | **49.0** |

**Table 3** – Semantic segmentation benchmarks **ADE20K** validation set with UPerNet (Xiao et al., 2018) and pre-trained ImageNet-1K backbone. All models use a crop size of $512 \times 512$.

GC ViT-B (52.9/45.8) backbone outperforms the counterpart with ConvNeXt-B (Liu et al., 2022) (52.7/45.6) by +0.2 and +0.2 in terms of box AP and mask AP, respectively.

## 4.3 ADE20K SEMANTIC SEGMENTATION RESULTS

We present semantic segmentation benchmarks on ADE20K dataset in Table 3. The models using pre-trained GC ViT-T (47.0), GC ViT-S (48.3) and GC ViT-B (49.0) backbones outperform counterpart models with pre-trained Twins-SVT-S (Chu et al., 2021a) (46.2), Twins-SVT-B (Chu et al., 2021a) (47.7) and Twins-SVT-L (Chu et al., 2021a) (48.8) by +0.8, +0.6 and +0.2 in terms of mIoU, respectively. In addition, models with GC ViT backbones significantly outperform counterparts with Swin Transformer backbones, hence demonstrating the effectiveness of the global self-attention.

## 5 ABLATION

### 5.1 COMPONENT-WISE ANALYSIS

As shown in Table 4, we study the role of each component in GC ViT model for classification, detection, instance and semantic segmentation. For simplicity, we start with Swin Transformer as the base model and progressively re-design the components to demonstrate their importance in improving the performance. Firstly, we remove the window shifting and predictably

| | ImageNet | COCO | | ADE20k |
|---|---|---|---|---|
| | top-1 | $AP^{box}$ | $AP^{mask}$ | mIoU |
| Swin-T | 81.3 | 50.4 | 43.7 | 44.5 |
| Swin-T w/o Window Shifting | 80.2 | 47.7 | 41.5 | 43.3 |
| + Reparam. (window, #blocks, ratio) | 81.9 | 50.5 | 43.7 | 45.0 |
| + GC ViT-T Stem | 82.2 | 50.7 | 43.9 | 45.2 |
| + GC ViT-T Down-sampler | 82.6 | 50.8 | 44.0 | 45.8 |
| **+ GC ViT-T Global Self-attention** | **83.4** | **51.6** | **44.6** | **47.0** |

**Table 4** – Ablation study on the effectiveness of various components in GC ViT on classification, detection and segmentation performance.

observe significant performance degradation across all tasks. Changing distribution of parameters to our design improves the performance by +1.7, +2.8, +2.2 and +1.7 in terms of accuracy, box AP, mask AP and mIoU. Such reparametrization includes changing the window size, MLP ratio, number of layers to name but a few. Adding the CNN-based stem of GC ViT to the model provides additional improvements of +0.3, +0.2, +0.2 and +0.2 in terms of accuracy, box AP, mask AP and mIoU. In addition, using our proposed downsampler further improves the accuracy, box AP, mask AP and mIoU by +0.4, +0.1, +0.1 and +0.3, respectively. The last two changes demonstrate the importance of convolutional inductive bias and capturing the inter-channel dependencies in our model. Finally, leveraging the proposed global self-attention improves the performance by by +0.8, +0.8, +0.6 and +1.2 in terms of accuracy, box AP, mask AP and mIoU. Hence, this validates the effectiveness of the

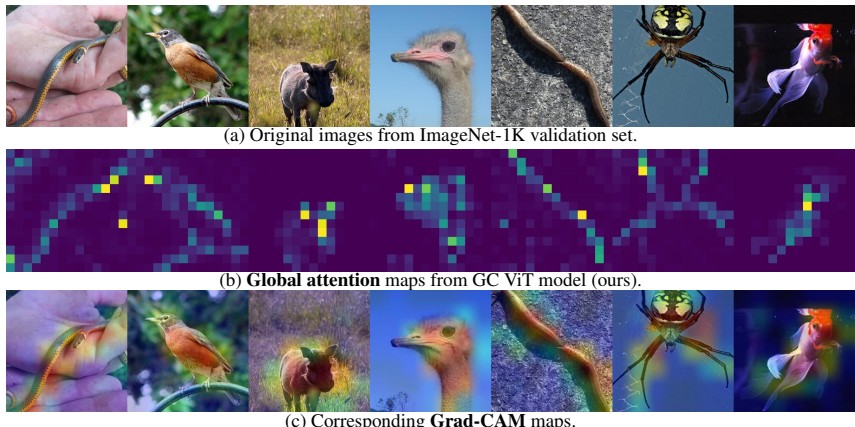

(a) Original images from ImageNet-1K validation set.

(b) **Global attention** maps from GC ViT model (ours).

(c) Corresponding **Grad-CAM** maps.

**Figure 7** – Visualization of : (a) input images (b) global self-attention maps from GC ViT-T model (c) corresponding Grad-CAM attention maps. Both short and long-range spatial dependencies are captured effectively. Please see the supplementary materials for illustration of learned global query token feature maps.

proposed global self-attention, in particular for downstream tasks with high resolution images such as semantic segmentation in which modeling long-range spatial dependencies is critical.

## 5.2 DOWNSAMPLER DESIGN

We studied the effectiveness of various downsampler blocks in Table 5. The simplest alternative to our design is a pair of convolutional and maxpooling layers. However, it results in a reduction of ImageNet Top-1 accuracy by -0.7. Patch merging is another variant which was introduced in Swin Transformers (Liu et al., 2021).

However, it reduces the accuracy by -0.5. Finally, our down-sampler which consists of a modified Fused-MBConv block and strided convolution and shows the best result. Importance of the former component is explained by the SE operation which boosts cross channel interaction while keeping number of parameters and FLOPs low. We conclude that our proposed down-sampler is essential to achieve high accuracy as it introduces convolutional inductive bias.

| Down-sampler | Architecture | Top-1 |
|---|---|---|
| Conv | Conv (s=1), Maxpool | 82.7 |
| Swin | Linear | 82.9 |
| **GC ViT** | Modified Fused-MBConv (s=2) | **83.4** |

**Table 5** – Ablation study on the effectiveness of down-sampler in GC ViT architecture on ImageNet Top-1 accuracy.

## 6 INTERPRETABILITY

To provide further insights on interpretability of the proposed global self-attention and query tokens, we demonstrate visualization of the learned attention and Grad-CAM (Selvaraju et al., 2017) maps in Fig. 7. The associated attention distributions uncovered by the global self-attention modules align with image semantics, and hence act as an informative source for local attention modules. In addition, corresponding Grad-CAM maps demonstrate accurate object localization with most intricate details.

## 7 CONCLUSION

We introduced a new vision transformer architecture named GC ViT that can efficiently capture global context by utilizing global query tokens and interact with local regions. Through extensive experiments we show SOTA benchmarks for image classification on ImageNet-1K dataset, surpassing CNN/ViT-based counterparts by large margin. We also consistently achieved SOTA for downstream tasks of detection, instance and semantic segmentation on MS COCO and ADE20K datasets.

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

# A  APPENDIX

# B  GC VIT MODEL CONFIGURATIONS

GC ViT model configurations are presented in Table S.1 describing the choice of internal hyper parameters to obtain models with various compute load and parameter number.

| | Output Size (Downs. Rate) | GC ViT-XT | GC ViT-T | GC ViT-S | GC ViT-B |
|---|---|---|---|---|---|
| Stem | 128×128 (2×) | Conv, C:64, S:2, LN
F-MBConv C:64 × 1 | Conv, C:64, S:2, LN
F-MBConv C:64 × 1 | Conv, C:96, S:2, LN
F-MBConv C:96 × 1 | Conv, C:128, S:2, LN
F-MBConv C:128 × 1 |
| Stage 1 | 56×56 (4×) | Conv, C:128, S:2, LN
LG-SA, C:64, head:2 × 3,
F-MBConv, C:128 | Conv, C:128, S:2, LN
LG-SA, C:64, head:2 × 3,
F-MBConv, C:128 | Conv, C:192, S:2, LN
LG-SA, C:96, head:3 × 3,
F-MBConv, C:192 | Conv, C:256, S:2, LN
LG-SA, C:128, head:4 × 3,
F-MBConv, C:256 |
| Stage 2 | 28×28 (8×) | Conv, C:256, S:2, LN
LG-SA, C:64, head:4 × 4,
F-MBConv, C:256 | Conv, C:256, S:2, LN
LG-SA, C:64, head:4 × 4,
F-MBConv, C:256 | Conv, C:384, S:2, LN
LG-SA, C:96, head:6 × 4,
F-MBConv, C:384 | Conv, C:512, S:2, LN
LG-SA, C:128, head:8 × 4,
F-MBConv, C:512 |
| Stage 3 | 14×14 (16×) | Conv, C:512, S:2, LN
LG-SA, C:64, head:8 × 6,
F-MBConv, C:512 | Conv, C:512, S:2, LN
LG-SA, C:64, head:8 × 19,
F-MBConv, C:512 | Conv, C:768, S:2, LN
LG-SA, C:96, head:12 × 19,
F-MBConv, C:768 | Conv, C:1024, S:2, LN
LG-SA, C:128, head:16 × 19,
F-MBConv, C:1024 |
| Stage 4 | 7×7 (32×) | Conv, C:1024, S:2, LN
LG-SA, C:64, head:16 × 5,
F-MBConv, C:1024 | Conv, C:1024, S:2, LN
LG-SA, C:64, head:16 × 5,
F-MBConv, C:1024 | Conv, C:1536, S:2, LN
LG-SA, C:96, head:24 × 5,
F-MBConv, C:1536 | Conv, C:2048, S:2, LN
LG-SA, C:128, head:32 × 5,
F-MBConv, C:2048 |

**Table S.1** – Architecture configurations for GC ViT. LG-SA and Conv denotes local, global self-attention and $3 \times 3$ convolutional layer, respectively. GC ViT-XT, GC ViT-T, GC ViT-S and GC ViT-B denote XTiny, Tiny, Small and Base variants, respectively.

# C  ABLATION

## C.1  GLOBAL QUERY

We performed ablation studies to validate the effectiveness of the proposed global query. Using the same architecture, instead of global query, we compute: (1) global key and value features and interact them with local query (2) global value features and interact it with local query and key. As shown in Table S.2, replacing global query may significantly impact the performance for image segmentation and downstream tasks such as object detection, instance segmentation and semantic segmentation.

| | ImageNet top-1 | COCO $AP^{box}$ | $AP^{mask}$ | ADE20k mIoU |
|---|---|---|---|---|
| w. Global KV | 82.5 | 49.9 | 41.3 | 44.6 |
| w. Global V | 82.7 | 50.8 | 42.4 | 45.1 |
| GC ViT-T | **83.4** | **51.6** | **44.6** | **47.0** |

**Table S.2** – Ablation study on the effectiveness of the proposed global query for classification, detection and segmentation.

## C.2  EMA AND BATCH SIZE

We also used used Exponential Moving Averages (EMA) and observed slight improvement in terms of ImageNet TOp-1 accuracy. Furthermore, the performance of the model across different batch sizes were stable as we did not observe significant changes. Table S.3 demonstrates the effect of EMA and batch size on the accuracy of a GCViT-T model.

| Model | Local Batch Size | Global Batch Size | EMA | Top-1 |
|---|---|---|---|---|
| GC ViT-T | 32 | 1024 | No | 83.37 |
| GC ViT-T | 128 | 4096 | No | 83.38 |
| GC ViT-T | 32 | 1024 | Yes | 83.39 |
| GC ViT-T | 128 | 4096 | Yes | 83.40 |

**Table S.3** – Ablation study on the effect of EMA and batch size on GC ViT-T ImageNet Top-1 accuracy.

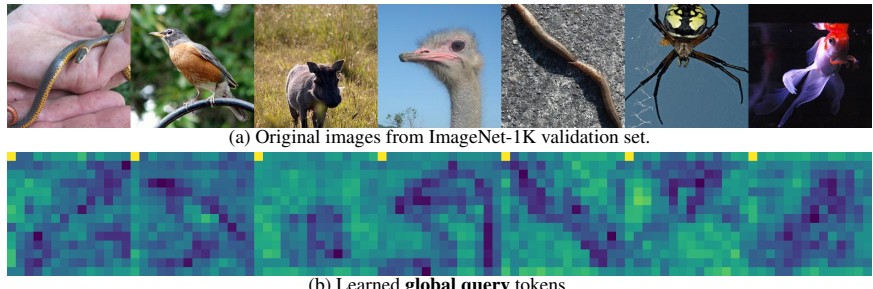
(a) Original images from ImageNet-1K validation set.

(b) Learned **global query** tokens.

**Figure S.1** – Visualization of : (a) input images (b) learned global query token feature maps.

## D    INTERPRETABILITY

In Fig. S.1, we illustrate the learned global query token maps and demonstrate their effectiveness in capturing long-range contextual representations from different image regions.

## E    TRAINING DETAILS

For image classification, GC ViT models were trained using four computational nodes with 32 NVIDIA A100 GPUs. The total training batch size is 1024 (32 per GPU) for GC ViT-S, GC ViT-B, GC ViT-L and 4096 (128 per GPU) for GC ViT-XXT, GC ViT-XT and GC ViT-T. On average, each model required 32 hours of training with the specified hyper-parameters as indicated in the paper. All classification models were trained using the `timm` package (Wightman, 2019). Object detection and instance segmentation models as well as semantic segmentation models were trained using one computational node with 8 NVIDIA A40 GPUs using a total batch size of 16, hence a batch size of 2 per GPU. Detection and instance segmentation models were trained using `mmdetection` (Chen et al., 2019) package and on average required 56 hours of training. Semantic segmentation models were trained using `mmsegmentation` (Contributors, 2020) package, and on average required 34 hours of training.

## F    COMPLEXITY ANALYSIS

Given an input feature map of $x \in \mathcal{R}^{H \times W \times C}$ at each stage with a window size of $h \times w$, the computational complexity of GC ViT is as follows

$$\mathcal{O}(\text{GC ViT}) = 2HW(2C^2 + hwC), \tag{7}$$

The efficient design of global query token generator and other components allows to maintain a similar computational complexity in comparison to Swin Transformer Liu et al. (2021) while being able to capture long-range information and achieve better higher accuracy for classification and downstream tasks such as detection and segmentation.

## G    COMPARISON TO OTHER GLOBAL SELF-ATTENTION MODULES

Other efforts such as EdgeViT (Pan et al., 2022) in computer vision and BigBird (Zaheer et al., 2020) in NLP have proposed global self-attention in their respective applications. In this section, we discuss the differences between the proposed global self-attention in GC ViT and these efforts.

| Model | Accuracy-Matched Frequency | Accuracy-Threshold-0.7 |
|---|---|---|
| GC ViT-XXT | 69.3 | 77.2 |
| GC ViT-XT | 71.3 | 78.8 |
| GC ViT-T | 73.1 | 80.5 |
| GC ViT-S | 73.8 | 80.7 |
| GC ViT-B | 74.4 | 81.1 |
| GC ViT-L | 74.5 | 81.5 |

**Table S.4** – Classiication benchmarks of GC ViT models on ImageNetV2 dataset.

| Model | Added Component | Top-1 |
|---|---|---|
| Swin-T | None | 81.3 |
| Swin-T | GC Module | 82.2 |
| Swin-S | None | 83.0 |
| Swin-S | GC Module | 83.7 |

**Table S.5** – Ablation study on the effectiveness of Global Context (GC) module in Swin Transformers architecture on ImageNet Top-1 accuracy.

**EdgeViT** : EdgeViT and GC ViT use completely different self-attention blocks. The EdgeViT uses a series of local aggregation (convolution), sparse attention and local propagation(depthwise convolution), whereas GC ViT only uses an interleaved pattern of local and global self-attention layers without convolution in order to compute self-attention. The proposed global sparse attention in EdgeViT and GCViT are competently different. EdgeViT samples representative tokens and only computes sparse self-attention between these representative tokens with reduced feature size. On the contrary, GC ViT computes self-attention between the global queries (not just the token) and local keys and values without any subsampling in their respective local regions. Furthermore, in EdgeViT, only subsampled representative tokens per region interact In the self-attention module; however, in GC ViT, the global queries interact with the entire local regions, instead of interacting with each other, and hence provide an effective mechanism for capturing both short and long-range spatial dependencies.

In addition, GC ViT generates global query tokens by using a series of modified Fused MB-Conv from the entire image and without subsampling. Note that the resolution of global query tokens are the same as local query and values. However, in EdgeViT: (A) the representative tokens are obtained per local window, not the entire image, and by subsampling and reducing the feature resolution. Hence, since generated tokens have a lower resolution compared to their respective local windows, this could result in loss of spatial information and impact the effectiveness of self-attention. Unlike EdgeViT, the downsampler in GCViT also benefits from modified Fused-MBConv blocks which allows for modeling cross channel interactions and impose more locality and convolutional inductive bias.

**BigBird** : Bigbird, which is primarily introduced for NLP applications with 1D inputs, has significant differences compared to GC ViT, which is proposed for computer vision with mainly 2D inputs. Firstly, BigBird uses a combination of random, window and global attention mechanisms, which is different from the proposed local and global self-attention scheme in GC ViT. In addition, BigBird does not have any specific mechanisms for extracting global tokens as the existing tokens or additional special tokens can be specified as global tokens. On the contrary, the global tokens in GC ViT are extracted by the proposed global query generator module which consists of a series of modified Fused MB-Conv blocks to extract contextual information from the entire input features. Lastly, BigBird employs a set of global tokens which attend to the entire input sequence; in this case, select global query, key and values attend to local query, key and value tensors. However, as opposed to this formulation, in GC ViT, the global query tokens attend to local key and value tokens in partitioned windows. This is due to the fact that attending to the entire input sequence, as done in BigBird, is not feasible considering the larger size of input features in computer vision.

## H    IMAGENETV2 BENCHMARKS

In Table S.4, we have evaluated the performance of GC ViT on ImageNetV2 dataset (**?**) to further measure its robustness. Specifically, we have used different sampling strategies of Matched Frequency and Threshold-0.7. These benchmarks demonstrate the competetive performance of GC ViT on ImageNetV2 dataset and validates its effectiveness in robustness and generalizability.

## I    EFFECT OF GLOBAL CONTEXT MODULE

In order to demonstrate the effectiveness of Global Context (GC) module, we use Swin Transformers as the base model and add our propoped GC module. In this analysis, we remove the window shifting operation from Swin Transformers, since GC module is capable of modeling cross-region interactions. As shown in Table S.5, addition of GC module improves the ImageNet Top-1 accuracy by $+0.9\%$ and $+0.7\%$ for Swin Transformers Tiny and Small variants respectively.

## J    IMAGENET CLASSIFICATION BENCHMARKS

In Table S.6, we provide a comprehensive benchmark in terms of Top-1 accuracy for the models that are only trained on ImageNet-1K (Deng et al., 2009) dataset, and without additional data.

**Table S.6** – Image classification benchmarks on **ImageNet-1K** dataset (Deng et al., 2009).

| Method | Param (M) | FLOPs (G) | Image Size | Top-1 (%) |
|---|---|---|---|---|
| ResMLP-S12 (Touvron et al., 2021a) | 15 | 3.0 | $224^2$ | 76.6 |
| PVT-v2-B1 (Wang et al., 2022) | 13 | 2.1 | $224^2$ | 78.7 |
| **GC ViT-XXT** | 12 | 2.1 | $224^2$ | 79.8 |
| EdgeViT-S (Pan et al., 2022) | 11 | 1.9 | $224^2$ | **81.0** |
| DeiT-Small/16 (Touvron et al., 2021b) | 22 | 4.6 | $224^2$ | 79.9 |
| T2T-ViT-14 (Yuan et al., 2021) | 22 | 5.2 | $224^2$ | 81.5 |
| **GC ViT-XT** | 20 | 2.6 | $224^2$ | **82.0** |
| ResNet50 (He et al., 2016) | 25 | 4.1 | $224^2$ | 76.1 |
| PVT-Small (Wang et al., 2021) | 24 | 3.8 | $224^2$ | 79.8 |
| Swin-T (Liu et al., 2021) | 29 | 4.5 | $224^2$ | 81.3 |
| CoAtNet-0 (Dai et al., 2021) | 25 | 4.2 | $224^2$ | 81.6 |
| Twins-SVT-S (Chu et al., 2021a) | 24 | 2.9 | $224^2$ | 81.7 |
| PVT-v2-B2 (Wang et al., 2022) | 25 | 4.0 | $224^2$ | 82.0 |
| ConvNeXt-T (Liu et al., 2022) | 29 | 4.5 | $224^2$ | 82.1 |
| Focal-T (Yang et al., 2021b) | 29 | 4.9 | $224^2$ | 82.2 |
| CSwin-T (Dong et al., 2022) | 23 | 4.3 | $224^2$ | 82.7 |
| **GC ViT-T** | 28 | 4.7 | $224^2$ | **83.4** |
| ResNet-101 (He et al., 2016) | 44 | 7.9 | $224^2$ | 77.4 |
| ResMLP-S24 (Touvron et al., 2021a) | 30 | 6.0 | $224^2$ | 79.3 |
| PVT-Medium (Wang et al., 2021) | 44 | 6.7 | $224^2$ | 81.2 |
| T2T-ViT-19 (Yuan et al., 2021) | 39 | 8.9 | $224^2$ | 81.9 |
| Twins-PCPVT-B (Chu et al., 2021a) | 44 | 6.7 | $224^2$ | 82.7 |
| Swin-S (Liu et al., 2021) | 50 | 8.7 | $224^2$ | 83.0 |
| Twins-SVT-B (Chu et al., 2021a) | 56 | 8.6 | $224^2$ | 83.2 |
| ConvNeXt-S (Liu et al., 2022) | 50 | 8.7 | $224^2$ | 83.1 |
| PVT-v2-B3 (Wang et al., 2022) | 45 | 6.9 | $224^2$ | 83.2 |
| CoAtNet-1 (Dai et al., 2021) | 42 | 8.4 | $224^2$ | 83.3 |
| Focal-S (Yang et al., 2021b) | 51 | 9.1 | $224^2$ | 83.5 |
| CSwin-S (Dong et al., 2022) | 35 | 6.9 | $224^2$ | 83.6 |
| **GC ViT-S** | 51 | 8.5 | $224^2$ | **83.9** |
| ResNet-152 (He et al., 2016) | 60 | 11.6 | $224^2$ | 78.3 |
| ViT-Base/16 (Dosovitskiy et al., 2020) | 86 | 17.6 | $224^2$ | 77.9 |
| ResMLP-B24 (Touvron et al., 2021a) | 116 | 23.0 | $224^2$ | 81.0 |
| PVT-Large (Wang et al., 2021) | 61 | 9.8 | $224^2$ | 81.7 |
| DeiT-Base/16 (Touvron et al., 2021b) | 86 | 17.6 | $224^2$ | 81.8 |
| CrossViT-B (Chen et al., 2021) | 104 | 21.2 | $224^2$ | 82.2 |
| T2T-ViT-24 (Yuan et al., 2021) | 64 | 14.1 | $224^2$ | 82.3 |
| CPVT-B (Chu et al., 2021b) | 88 | 17.6 | $224^2$ | 82.3 |
| Twins-PCPVT-L (Chu et al., 2021a) | 61 | 9.8 | $224^2$ | 83.1 |
| Swin-B (Liu et al., 2021) | 88 | 15.4 | $224^2$ | 83.3 |
| CoAtNet-2 (Dai et al., 2021) | 42 | 8.4 | $224^2$ | 83.3 |
| PVT-v2-B4 (Wang et al., 2022) | 62 | 10.1 | $224^2$ | 83.6 |
| Twins-SVT-L (Chu et al., 2021a) | 99 | 15.1 | $224^2$ | 83.7 |
| ConvNeXt-B (Liu et al., 2022) | 89 | 15.4 | $224^2$ | 83.8 |
| Focal-B (Yang et al., 2021b) | 90 | 16.0 | $224^2$ | 83.8 |
| PVT-v2-B5 (Wang et al., 2022) | 82 | 11.8 | $224^2$ | 83.8 |
| CSwin-B (Dong et al., 2022) | 78 | 15.0 | $224^2$ | 84.2 |
| BoTNet (Dong et al., 2022) | 79 | 19.3 | $256^2$ | 84.2 |
| **GC ViT-B** | 90 | 14.8 | $224^2$ | **84.4** |
| ConvNeXt-L (Liu et al., 2022) | 198 | 34.4 | $224^2$ | 84.3 |
| CoAtNet-3 (Dai et al., 2021) | 168 | 34.7 | $224^2$ | 84.5 |
| **GC ViT-L** | 201 | 32.6 | $224^2$ | **84.6** |

