# OpenReview forum: "Global Context Vision Transformers"
_ICLR.cc/2023/Conference — Submitted to ICLR 2023_

### Official Review · Reviewer_V9pP · 2022-10-20

**Confidence:** 5
**Correctness:** 3
**Technical Novelty And Significance:** 2
**Empirical Novelty And Significance:** 2
**Recommendation:** 6

**Clarity, Quality, Novelty And Reproducibility:**

> Clarity

Good.

> Quality

Good.

> Novelty

Good.

> Reproducibility

No code is available.

**Details Of Ethics Concerns:**

No further concerns.

**Strength And Weaknesses:**

> Strengths

✅ The presented idea is very simple and easy to follow.

✅ The overall writing is of high quality, such as Figure 2, Figure 3, Figure 4, Figure 5, and Figure 6 is all of GOOD quality and help the readers to understand the key
idea easily.

✅ The authors provide rich details and conduct rich experiments to verify the effectiveness of the proposed approach.

> Weaknesses

❎ The experimental results are relatively WEAK and the authors are encouraged to verify the advantages over Swin-T based on the very recent object detection and segmentation frameworks
such as DINO-DETR[1], H-DETR[2], and Mask2Former[3]. For example, Mask2Former+Swin-B (pre-trained on ImageNet1K) achieves PQ=55.1, AP=46.7, and mIoU=52.4 on COCO panoptic segmentation, COCO instance
segmentation, and ADE20K semantic segmentation tasks respectively. I would like to expect the authors to present stronger results and release the related code upon acceptance.

[1] https://github.com/IDEA-Research/DINO

[2] https://github.com/HDETR/H-Deformable-DETR

[3] https://github.com/facebookresearch/Mask2Former

❎ According to Table 1 & Table 2, the proposed GC Vit backbones require more parameters and GFLOPs than the previous approaches such as CSWin and ConvNeXt. For example, GC ViT-B requires 1018 GFLOPs vs. ConvNeXt-B requires 964 GFLOPs and GC ViT-B only gains 0.2 measured by AP^box.

❎ The authors are encouraged to verify the advantages of the proposed backbone under larger model scales. Specifically, the authors should construct a GC ViT-L and GC ViT-H instead of only comparing the models under moderate scales as scaling up is one of the most important aspects of an important backbone design.

**Summary Of The Paper:**

This paper proposes a novel global context vision transformer built with a simple combination of global context self-attention modules and local self-attention modules to model
both long-range and short-range interactions. Besides, the authors propose to model the inter-channel dependencies with an inverted residual block.
The proposed GC ViT shows encouraging results across image classification, object detection, instance segmentation, and semantic segmentation tasks.


**Summary Of The Review:**

Designing new versatile architecture is always very important for the whole computer vision community. The overall quality is OK but the results are relatively WEAK to support the claim that the proposed method really outperforms the previous SoTA methods such as Swin-Transformer.
Please carefully address the above-listed weaknesses. I will increase the ratings if the authors could well address these concerns.

---

> ### Author Response · Authors · 2022-11-18
> **Response to Reviewer V9pP**
>
> We are sincerely appreciative of the reviewers' efforts as well as the feedback and suggestions for the paper. We are glad to see
> that the reviewer finds the design of our experiments to be detailed enough to verify the effectiveness of the proposed approach, and the overall presentation of our work to be high quality. We have provided detailed feedback to the reviewer’s comments in the following.
>
> > **Release of the code**
>
> We appreciate the reviewer’s comment for the release of the code. As part of this rebuttal, we have anonymously released the code to the reviewers. In addition, our code will be publicly released upon the acceptance of this work.
>
> > **The experimental results are weak for object detection and segmentation over H-DETR, etc.**
>
> We thank the reviewer for their comment. In order to demonstrate the effectiveness of our proposed model as a viable backbone for different downstream tasks, we strictly followed the exact training configurations as those of **ConvNeXt** and **Swin Transformers** for fair comparison. Hence, we would like to emphasize that our goal for downstream tasks was not to outperform other SOTA methods (e.g. **H-DETR**), and as a result we used commonly used **MaskRCNN** and **UPerNet** as detection/segmentation heads. In our benchmarks, our model significantly outperforms both **ConvNeXt** and **Swin Transformers** for various model sizes. For instance, **GC ViT-T** outperforms **Swin-T** and **ConvNeXt-T** by **+1.7** and **+1.9** in terms of box AP and **+1.5** and **+1.6** in terms of mask AP, respectively.
>
> In addition, we have taken the reviewer’s comment into consideration for evaluation of our models with SOTA methods, and we have trained a **H-Deformable-DETR** network with **GC ViT-T** backbone on MS COCO dataset for object detection. Our model achieves an AP of **53.4** with the training configuration of the official repository with decay of **1e-4** and **36 epochs**. For comparison, a similar model with the same training configuration and **Swin-T** backbone achieves an AP of **53.2**.  Hence, this experiment validates the effectiveness of our model in finetuning downstream tasks with more advanced networks. We will continue to add more advanced models to our detection and segmentation repository for the public release of our code.
>
> > **GC ViT backbones require more parameters and GFLOPs**
>
> We appreciate the reviewer's comment. For semantic segmentation, the UPerNet model with **GC ViT-B** has **125M** parameters and achieves a **mIOU** of **49.0**. On the other hand, the closest competitor with **Twins-SVT-L**  backbone has **133M** parameters and achieves a **mIOU** of **48.8**. Hence, the model with **GC ViT-B** backone has **8M** less number of parameters, and still outperforms the corresponding model with **Twins-SVT-L** by **+0.2** in terms of **mIOU**.
>
> For object detection, the **Cascade Mask-RNN** with **GC ViT-B** backbone has  **5.3%** more FLOPs than the corresponding model with **ConvNeXt-B** backbone. However, both models have the same number of parameters with **146M**. Considering the number of parameters in both of the models, and the small increase in number of FLOPs for **GC ViT-B**, we believe that  both models are very comparable in terms of their computational requirements.
>
> We would like to also note that, the **Cascade Mask-RCNN** with **Swin-B** backbone has more FLOPs than the corresponding model with **ConvNeXt-B** backbone and is closer to the model with **GC ViT-B** backbone in this regard. However, the model with **GC ViT-B**  backbone significantly outperform the corresponding **Swin-B** model by **+1.0** in terms of box AP, hence demonstrating the effectiveness of our model when compared to a model with larger number of GFLOPs than **ConvNeXt**.
>
>
> > **Constructing Larger GC ViT variants and scaling up the model**
>
> We thank the reviewer for their comment as scaling up the model/dataset size is an important aspect for creating novel general purpose backbones. In Table 1, we have reported numbers for **GC ViT-L** for ImageNet-1K dataset. Specifically, **GC ViT-L** with **201M** parameters and **32.6 GLOPs**, achieves a Top-1 accuracy of **84.6%**, outperforming **ConvNeXt-L** by **0.3%**.
>
> We have also considered the reviewers request for scaling up the model and dataset and pre-traiend **GC ViT-L** on **ImageNet-21K** dataset. Finetuning this model on **ImageNet-1K** dataset achieves a Top-1 accuracy of **86.5%**. We have also constructed **GC ViT-H** model with **803M** parameters and **124.2GFLOPs**. We will add pre-trained GC ViT-H model weights (ImageNet-1K, ImageNet-21K) to our public repository upon its release.
>
> If our responses have properly addressed the reviewer's concerns, we would like to kindly request the reviewer to consider raising their scores accordingly. Otherwise, we will be happy to further discuss.

---

### Official Review · Reviewer_Fuf9 · 2022-10-25

**Confidence:** 3
**Correctness:** 3
**Technical Novelty And Significance:** 2
**Empirical Novelty And Significance:** 3
**Recommendation:** 6

**Clarity, Quality, Novelty And Reproducibility:**

There are no problems with Clarify, Quality and Novelty.
However, since the paper is based on plain empirical/experimental findings. The released code will be critical in verifying the reproducibility of the paper.

**Strength And Weaknesses:**

Strength:
1. the idea is simple and straightforward, from the experiment it seems the improvement is reasonable.

Weakness:
1. The comparison of GC-VIT to other models clearly shows the quantitative improvements in Tab.1. In addition, the ablation study shows the proposed module could be quite plug-able into existing models. It will be more appealing to see how GC module improves existing models by simply plug them into them. This will inevitably add #param  and FLOPS, but it gives more intuitive evidence on the effectiveness.

**Summary Of The Paper:**

This paper introduced GC ViT that can efficiently capture global context by utilizing global query tokens and interact with local regions.
This paper demonstrate through experiments that this simple modification on traditional ViTs could get SOTA results for image classification on ImageNet-1K dataset and downstream tasks.

**Summary Of The Review:**

Overall this paper proposed a simple and yet effective module that could add reasonable improvement to current ViT. This work could be useful to the community once the code is released for verification/comparison.

---

> ### Author Response · Authors · 2022-11-17
> **Response to Reviewer Fuf9**
>
> We would like to sincerely appreciate the reviewer’s efforts in reviewing our work and providing constructive feedback. We appreciate the fact that the reviewer has found our work to be straightforward with clear quantitative improvements over previous efforts. We have provided answers to the reviewer’s comments in the following.
>
>
> > **Release of the code**
>
> We thank the reviewer for this request. Firstly, we have anonymously and exclusively released the code to the reviewers for their perusal. In addition, we agree with the reviewer that our code repository will be a useful resource for the research community, and it will enable further verification and comparison. As such our code and pre-trained models will be publicly released.
>
> > **Improvement of other models by plug-and-play GC module**
>
> This is a great insight which provides further opportunity to benefit from our proposed **GC module** across a variety of models. As such, we have used **Swin Transformer** as the basis, and used our proposed GC module (i.e. global query, global attention, etc.) within the various variants of their architecture. For this comparison, we have removed window shifting as the proposed global query tokens can sufficiently model long-range spatial dependencies and capture cross-window interactions.
>
> | Model   |Added Component |Top-1|
> |-------------------|--------|--------|
> | Swin-Tiny  |None| 81.3|
> | Swin-Tiny| GC Module|82.2|
> | Swin-Small  | None|83.0|
> | Swin-Small| GC Module|83.7|
>
> We observe that the addition of our proposed GC module improves the ImageNet Top-1 accuracy by **+0.9%** and **+0.7%** for **Swin Transformers** **Tiny** and **Small** variants respectively. Note that we did not make any additional architecture changes, such as our proposed downsampler which would further improve the accuracy. We have added this analysis to the supplementary materials of the revised manuscript.

---

### Official Review · Reviewer_t1Sn · 2022-10-27

**Confidence:** 5
**Correctness:** 1
**Technical Novelty And Significance:** 1
**Empirical Novelty And Significance:** 1
**Recommendation:** 1

**Clarity, Quality, Novelty And Reproducibility:**

The paper is well written and easy to follow. Unfortunately, the claims did not seem to be supported and the novelty seems very marginal. The paper seems to be reproducible

**Strength And Weaknesses:**

Strength
- The paper is well written and easy to follow
- There is evaluation on different tasks

Weaknesses:
- Missing comparaison and discussion:

a) The global query tokens claim: The proposed global query token behaviour seem to be quite similar to the attention pattern mechanism in the BigBird paper[1] and also the local global attention in the paper EdgeViT[2]. It is necessary to discuss the contribution of the proposed method to these two papers. Because at present the contribution of the paper on this aspect is not clear.

[1] Zaheer et al., Big Bird: Transformers for Longer Sequences
[2] Pan et al., EdgeViTs: Competing Light-weight CNNs on Mobile Devices with Vision Transformers

b) The compute and parameter-optimized hierarchical ViT and SOTA claim: in Figure 1 and Table 1: Indeed, the EdgeViT paper exploit similar ideas (local-global attention, Hierachical architecture) and seem to have better FLOPs-accuracy and Parameter accuracy trade-off (EdgeViT-S: 11.1M params, 1.9G FLOPs, 81.0 top-1 on ImageNet vs GC ViT-XXT 12M params, 2.1G FLOPs, 79.8 top-1 on ImageNet)  It is therefore necessary to add this architecture to the comparison. Moreover, the paper mention EfficientNetV2[1] because GC ViT exploit some similar blocks and seem also better/similar for the FLOPs-accuracy and parameter accuracy trade-off (EfficientNetV2-M 54M params, 24G flops, 85.1 top-1 on ImageNet  vs GC ViT-L  201M params, 32.6 G FLOPs, 84.6 top-1 ImageNet). So it's important to add also EfficinetNet-v2 in the comparaison.

Given these elements it is probably best to down tone the state-of-the art claims in the paper. Because it is not clear which state of the art it is.

[1] Tan et al., EfficientNetV2: Smaller Models and Faster Training

b) The efficient CNN-like token generator: Many papers have proposed convolution blocks for token generation like LeViT[1] or XCiT[2].
There is no discussion or comparison of performance with the previous approach. It is therefore impossible to evaluate the contribution of the paper on this point.

[1] Graham et al., LeViT: a Vision Transformer in ConvNet's Clothing for Faster Inference
[2] El-Nouby et al., XCiT: Cross-Covariance Image Transformers

- Overfitting evaluation: For the ImageNet dataset there is no separate test and validation set so it's important to evaluate the level of overfitting by doing evaluation of the models on ImageNet-v2[1].

[1] Recht et al., are we done with ImageNet?

- Missing metrics: There is only params and FLOPs in the different table. It's important to measure other trade-off like Latency and memory consumption. Indeed, the proposed architecture use MBconv which are known to be good on trade-off FLOPs accuracy and parameter accuracy.




**Summary Of The Paper:**

The paper proposed a new hirerchical transformers architecture. The paper revisit the attention mechanism and the downsampling layers.
The claim of the paper are the following:

• We introduce a compute and parameter-optimized hierarchical ViT with reparametrization of the
design space (e.g., embedding dimension, number of heads, MLP ratio).

• We design an efficient CNN-like token generator that encodes spatial features at different resolutions
for global query representations.

• We propose global query tokens that can effectively capture contextual information in an efficient
manner and model both local and global interactions.

• We introduce a parameter-efficient downsampling module with modified Fused MB-Conv blocks
that not only integrates inductive bias but also enables the modeling of inter-channel dependencies.

• We demonstrate new SOTA benchmarks for : (1) ImageNet classification with Pareto fronts on
ImageNet-1K for model size and FLOPs (see Fig. 1), and (2) downstream tasks such as detection,
instance segmentation and semantic segmentation on MS COCO and ADE20K, respectively.


**Summary Of The Review:**

The idea of the paper is interesting but unfortunately the contribution does not seem to be justified experimentally. The paper seems to be incremental. There is a lack of discussion and comparison with very similar approaches in the literature.


====== Post rebuttal ====

The rebuttal raised a very important concern:

The authors report maximum accuracy during training by evaluating at each epoch in order to maximise performance on the ImageNet-1k validation set (there is no separate test set for ImageNet). Although the authors argue that it is a common practice, there is no evidence in the mentioned papers that this is the approach used. Furthermore I consider this approach to be detrimental to the community as it puts scientific understanding behind the optimal performance.

On the 3 logs provided the impact of this practice is important on the log with the bigger model +0.2% and impact the conclusion. The authors no longer have the other logs of their experiments, especially those of the ablation, so it is impossible to know the impact of the method. There is no multi-seed evaluation that shows that 0.1% on ImageNet-v2 is significant although it is claimed in the rebuttal.

In the current state the experiments are mainly designed to optimise the performance of the GC ViT architecture at the expense of an understanding of the impact of each component of the proposed method.

So I will lower my score.

---

> ### Author Response · Authors · 2022-11-17
> **Response to Reviewer t1Sn - 1**
>
> We provide detailed responses to the reviewer’s comments in the following.
>
> > **GC ViT has similar ideas to Edge ViT. Global query token is similar to Big Bird and EdgeViT.**
>
>
> We respectfully disagree with the reviewer as Big Bird (NLP) and EdgeViT (CV) have entirely different attention mechanisms than GC ViT. In addition, GC ViT and EdgeViT have many differences and follow different ideas. We provide a detailed discussion regarding these differences in the following.
>
>
>
> **EdgeViT**: We discuss many differences between the GC ViT and EdgeViT in the following.
>
> **1) Attention mechanism + Block design**
>
>
> EdgeViT and GC ViT use completely different self-attention blocks. The EdgeViT uses a series of local aggregation (convolution), sparse attention and local propagation(depthwise convolution), whereas GC ViT only uses an interleaved pattern of local and global self-attention layers without convolution in order to compute self-attention. The proposed global sparse attention in EdgeViT and GCViT are competently different. EdgeViT samples representative tokens and only computes sparse self-attention between these representative tokens with reduced feature size. On the contrary, GC ViT computes self-attention between the global queries (not just the token) and local keys and values without any subsampling in their respective local regions. Furthermore, in EdgeViT, only subsampled representative tokens per region interact In the self-attention module; however, in GC ViT, the global queries interact with the entire local regions, instead of interacting with each other, and hence provide an effective mechanism for capturing both short and long-range spatial dependencies.
>
>
> **2) Global Token Generation**
>
>
> GC ViT generates global query tokens by using a series of modified Fused MB-Conv from the entire image and without subsampling. Note that the resolution of global query tokens are the same as local query and values. However, in EdgeViT: (A) the representative tokens are obtained per local window, not the entire image, and by subsampling and reducing the feature resolution. Hence, since generated tokens have a lower resolution compared to their respective local windows, this could result in loss of spatial information and impact the effectiveness of self-attention.
>
> **3) Downsampler**
>
> Unlike EdgeViT, the downsampler in GCViT also benefits from modified Fused-MBConv blocks which allows for modeling cross channel interactions and impose more locality and convolutional inductive bias.
>
>
> We have released our code repository to reviewers and ACs. In addition, the public code repository of EdgeViT is available in the following. Hence, direct comparisons of GC ViT and EdgeViT repositories can further elucidate the many differences between these efforts:
>
> https://github.com/saic-fi/edgevit
>
>
> **BigBird**: Bigbird, which is primarily introduced for NLP applications with 1D inputs, has significant differences compared to GC ViT, which is proposed for computer vision with mainly 2D inputs. These differences are listed in the following.
>
> **1)  Pattern of Attention Mechanism**
>
> BigBird uses a combination of random, window and global attention mechanisms, which is different from our proposed local and global self-attention scheme
>
> **2) Selection/Generation of Global Token**
>
> BigBird does not have any specific mechanisms for extracting global tokens as the existing tokens or additional special tokens can be specified as global tokens. On the contrary, the global tokens in GC ViT are extracted by the proposed global query generator module which consists of a series of modified Fused MB-Conv blocks to extract contextual information from the entire input features.
>
> **3) Global Attention Mechanism**
>
> BigBird employs a set of global tokens which attend to the entire input sequence; in this case, select global query, key and values attend to local query, key and value tensors. However, as opposed to this formulation, in GC ViT, the global query tokens attend to local key and value tokens in partitioned windows. This is due to the fact that attending to the entire input sequence, as done in BigBird, is not feasible considering the larger size of input features in computer vision.
>
>
> Per reviewer’s recommendation, we have cited EdgeViT and updated the revised manuscript and its supplementary materials to reflect the differences between these efforts.

---

> > ### Author Response · Authors · 2022-11-17
> > **Response to Reviewer t1Sn - 2**
> >
> >
> >
> > > **EdgeViT has a better FLOPs-accuracy and Parameter accuracy trade-off than GC ViT**
> >
> > We appreciate the reviewer’s comment regarding EdgeViT and GC ViT. However, we believe that EdgeViT and GCViT cannot be directly compared as each of them have been designed for a different purpose. EdgeViT is specifically designed for light-weight mobile devices whereas GC ViT is a general backbone which covers a spectrum of models with different Params/Flops. The only models that could be directly compared are **EdgeViT-S** (**11.1M**/**1.9G**) with **GC ViT-XXT**(**12M**,**2.1G**), which is the smallest GC ViT model. However, a comparison between a single model cannot justify the argument regarding FLOP/Parameter-accuracy trade-off as both methods have substantially different model sizes, and are designed for different applications.
> >
> >
> > We also believe that performance of GC ViT should be measured against comparable approaches with similar model sizes/FLOPS that are designed as general computer vision backbones and include a variety of variants with different sizes – we were not able to compare to any other EdgeViT other than one model. Specifically,  **GC ViT-T**, **GC ViT-S** and **GC ViT-B** outperform comparable mainstream models such as Swin Transformers variants by **+2.1%**, **+0.8%**, and **+1.1%**, and  **ConvNeXt** variants by **+1.3%**, **+0.7%** and **+0.6%** respectively.
> >
> >
> > > **It is necessary to add EdgeViT for the comparison**
> >
> > We thank the reviewer for this suggestion. We have added EdgeViT to Table 1 and Related Work section in the revised manuscript.
> >
> > > **EfficientNetV2 achieves better FLOPs/ Parameter-accuracy and need to be added**
> >
> > We appreciate the reviewer's comment regarding EfficientNetV2. However, the numbers reported in the EfficientNetV2 paper are not directly comparable to our work. This is due to the fact that their model has been trained for **350 epochs** as opposed to the **300 epochs** for our model and those listed in **Table 1**. Moreover, their training configuration uses **advanced tricks** such as **progressive learning** which uses different image resolutions and augmentation during training.
> >
> > Regarding the FLOPs/ Parameter-accuracy trade-off comparison, we believe that a single comparison between **EfficientNetV2-M** and **GC ViT-L**, which are trained under different configurations, may not be conclusive. This argument can also be extended for other model variants. For instance, **GC ViT-S** has **3.5%** less number of FLOPs compared to **EfficientNetV2-M**, while both models achieve the same Top-1 accuracy of **83.9%** on ImageNet. However, despite the seemingly better FLOPs-accuracy trade-off for  **GC ViT-S** model in this instance, different training configurations of these models baffle their direct comparison.
> >
> >
> > > **Token generation of LeViT and XCiT**
> >
> > We thank the reviewer for this comment. However, XCiT and LeViT convert an image to tokens with a patch embedding layer consisting of a stack of convolutional layers, whereas token generation in GC ViT denotes extraction of global queries  from intermediate feature maps, as opposed to input images, and by using modified Fused MB-Conv layers. This design choice in leveraging Fused MB-Conv layers results in an efficient way of generating global query tokens.
> >
> > > **Model Evaluation on ImageNet-v2**
> >
> > We thank the reviewer for this suggestion as it can further bolster the analysis of our model’s performance. We have evaluated different variations of GC ViT on **ImageNet-v2** dataset using two different approaches of Matched Frequency and Threshold-0.7 as the sampling strategies. The experimental results, as listed below, demonstrate a competitive performance on ImageNet-v2 dataset, hence validating the robustness and generalizability of the proposed GC ViT model. We have also added these benchmarks to the supplementary materials of the revised manuscript.
> >
> > | Model   | ImageNet-1K | ImageNet-v2 (MF) | ImageNet-v2 (Th-0.7) |
> > |-------------------|----------------|----------------|--------|
> > | GCViT-XXT |79.8  |69.3 | 77.2|
> > | GCViT-XTiny |82.0  | 71.3 | 78.8 |
> > | GCViT-Tiny|83.4  |73.1 | 80.5 |
> >  |GCViT-Small |83.9 |73.8  | 80.6 |
> >  |GCViT-Base |84.4 |74.4  | 81.1 |
> >  |GCViT-Large |**84.8** |**74.9**  | **81.9** |
> >
> > MF and Th-0.7 denote Matched Frequency and Threshold-0.7 samplers, respectively.

---

> > > ### Comment · Reviewer_t1Sn · 2022-11-18
> > > **Additional remarks concerning the logs**
> > >
> > > Dear Authors,
> > >
> > > Thank you for providing the code and logs of the ImageNet experiments.
> > >
> > > I have additional remarks concerning the logs:
> > >
> > > 1) In the paper we have: "with the AdamW (Loshchilov& Hutter, 2017) optimizer for 300 epochs with an initial learning rate of 0.001" but according to the logs it seems that the training is done during 320 epochs. What explains the difference?
> > >
> > > 2) According to the log of GC ViT-L it seems that the model is evaluated every epoch on ImageNet val and that it is the max accuracy during the training that is reported and not the final accuracy. This does not seem to be a good practice because it favors overfitting and makes the comparison with other approaches more complex. It would be interesting to have the final performance for all the results and not the maximum if it is not already the case.
> > > Indeed, this can change the conclusion for example from the provided logs it seems that GC ViT-B and GC ViT-L have the same performance so is not clear if the proposed method scale or not.
> > >
> > > If possible, can you provide all the logs of the different experiments (ablation, ADE20K, COCO) ?

---

> > > > ### Author Response · Authors · 2022-11-19
> > > > **Response to Reviewer t1Sn Concerning the Logs**
> > > >
> > > > We thank the reviewer for their comments and engaging in further discussion regarding our work. We hope that our previous detailed answers have properly addressed the the reviewer's main concerns (e.g. novelty, commonalities with EdgeViT) as the reviewer has not provided any additional feedback or discussion regarding these concerns.
> > > >
> > > > Below please see our detailed responses regarding the additional concerns as raised by the reviewer.
> > > >
> > > > > **What explains the difference in number of epochs**
> > > >
> > > > We thank the reviewer for their comment. In **Sec.4**, we have detailed all training details. Specifically, we have indicated **"300 epochs with an initial learning rate of 0.001, weight decay of 0.05, cosine decay scheduler and 20 warm-up and cool-down epochs"**. Hence the model is trained for an additional **20 cool-down epochs** which is consistent with all of the provided training logs. The cool-down epochs are used as part of different learning rate schedulers (e.g. cosine) to reduce the learning rate to its minimum value. Using the cool-down epochs is a common practice among previous efforts for training on the ImageNet-1K dataset. Furthermore, according to our benchmarks and other comparable models, such small number of epochs at the end of the training with the lowest learning rate does not have a considerable impact on the final accuracy.
> > > >
> > > > > **GC ViT-L evaluated every epoch and it may complicate comparisons**
> > > >
> > > > We respectfully disagree with the reviewer. Considering the large size of ImageNet-1K dataset, it is indeed a common practice to evaluate models at the end of every training epoch and keep track of the accuracy. For instance please refer to Swin Transformers and ConvNext official repositories. For our benchmarks on ImageNet-1K dataset, the best and final accuracy for all models are very similar, as the variations happen only on the 2nd decimal, and as a result we only report accuracy with the 1st decimal to mitigate these issues ( please see Table 1).
> > > >
> > > > Moreover, previously, the reviewer indicated that **"For the ImageNet dataset there is no separate test and validation set so it's important to evaluate the level of overfitting by doing evaluation of the models on ImageNet-v2"**. Hence according to the reviewer's previous comment, the best way to evaluate the level of overfitting is to employ another dataset. We have considered this request and reported the following benchmarks on ImageNet-v2 dataset which verifies the generalizability of our model across different model variants without overfitting:
> > > >
> > > >
> > > > | Model   | ImageNet-1K | ImageNet-v2 (MF) | ImageNet-v2 (Th-0.7) |
> > > > |-------------------|----------------|----------------|--------|
> > > > | GCViT-XXT |79.8  |69.3 | 77.2|
> > > > | GCViT-XTiny |82.0  | 71.3 | 78.8 |
> > > > | GCViT-Tiny|83.4  |73.1 | 80.5 |
> > > >  |GCViT-Small |83.9 |73.8  | 80.6 |
> > > >  |GCViT-Base |84.4 |74.4  | 81.1 |
> > > >  |GCViT-Large |**84.8** |**74.9**  | **81.9** |
> > > >
> > > > Considering our answers in the above, the conclusion regarding our model performance remains the same, and it outperforms previous SOTA approaches such as Swin Transformer and ConvNeXt, sometimes by large margins.
> > > >
> > > > > **GC ViT-L may not scale**
> > > >
> > > > On **ImageNet-1K** dataset, **GC ViT-L** outperforms the comparable **ConvNeXt-L** model by **+0.3%** in terms of Top-1 accuracy. In addition, it outperforms **GC ViT-B** by **+0.4%**. Considering the reviewer's argument, we refer to our benchmarks on **ImageNetv2 dataset** as shown above, in which **GC ViT-L** outperforms **GC ViT-B** by **+0.5%** and **+0.8%** in terms of accuracy with Matched Frequency and Threshold-0.7 samplers, respectively. Hence, this refutes the claim of overfitting to the validation set, since **GC ViT-L** consistently outperforms **GC ViT-B** on different evaluation sets. In addition, we have taken the suggestion made by **Reviewer V9pP** into consideration and scaled up **GC ViT-L** in terms of the dataset size. Pre-training on **ImageNet-21K** and fine-tunning on **ImageNet-1K** datasets yields a Top-1 accuracy of **86.5%**. Hence this validates the scalability of **GC ViT-L** model in terms of both model and dataset sizes. Unfortunately, we don't have the requested logs for all experiments at our disposal.

---

> > > > > ### Comment · Reviewer_t1Sn · 2022-11-19
> > > > > **Response: Additional remarks concerning the logs**
> > > > >
> > > > > Dear Authors,
> > > > >
> > > > > Thank you for your answer and the clarification this is very helpful, I have new questions following your response:
> > > > >
> > > > > - *"we refer to our benchmarks on ImageNetv2 dataset as shown above, in which GC ViT-L outperforms GC ViT-B by +0.1%"* according to Wightman et al.[1] 0.1% on ImageNetv2 is not significant, do you have a multi-seed evaluation to measure the standard deviation of the results?
> > > > >
> > > > > [1] Wightman et al., ResNet strikes back: An improved training procedure in timm
> > > > >
> > > > > - For the ImagNet-v2 results is it the max accuracy during training or the performance of the best model on ImageNet-val?
> > > > >
> > > > > - *"the best and final accuracy for all models are very similar, as the variations happen only on the 2nd decimal"* but for GC ViT-L in Table 1 we have 84.6 and in the logs we have 84.41 so the gap seem to be 0.19%.
> > > > >
> > > > > - *"Unfortunately, we don't have the requested logs for all experiments at our disposal."* if possible can you provide those that are at your disposal ?

---

> > > > > > ### Author Response · Authors · 2022-11-20
> > > > > > **Response to Additional Remarks Concerning Logs**
> > > > > >
> > > > > > We thank the reviewer for additional remarks and comments. Please see our detailed answers in the following.
> > > > > >
> > > > > > > **0.1% on ImageNetv2 is not significant**
> > > > > >
> > > > > > We believe that the conclusion by Wightman et al.[1] is specific to experiments concerning improved training procedure for ResNet. In addition, the performance on ImageNetv2 is largely influenced by the sampling strategy [2]. To address the concerns regarding the performance of **GC ViT-Large** model, we have re-trained it on ImageNet-1K dataset. Our results show that **GC ViT-Large** outperforms **GCViT-Base** by **0.5%** and **0.8%** in terms of accuracy with Matched Frequency and Threshold-0.7 samplers, respectively. Hence, this validates the scalability and generalizability of **GC ViT-Large**. We present these results in the following table.
> > > > > >
> > > > > > | Model   | Acc. (Matched Frequency) | Acc. (Threshold-0.7) |
> > > > > > |-------------------|----------------|--------|
> > > > > >  |GC ViT-Base|74.4  | 81.1 |
> > > > > >  |**GC ViT-Large** |**74.9**  | **81.9** |
> > > > > >
> > > > > > In addition to ImageNetV2, we have decided to use a second external dataset to further demonstrate the robustness of our proposed GC ViT model. Specifically, we have resorted to using **ImageNet-A dataset** [3] which contains real-worlds and naturally occurring images. As a results, this dataset is more challenging. In the table below, we present the performance of **GC ViT-Base** and **GC ViT-Large** along with **Swin Transformer** and **ConvNext**.
> > > > > >
> > > > > > | Model   | Acc. |
> > > > > > |-------------------|----------------|
> > > > > > | Swin-Base  |35.8 |
> > > > > > | ConvNeXt-Base  |36.7 |
> > > > > > | GC ViT-Base  |42.6 |
> > > > > > | **GC ViT-Large**  |**45.8** |
> > > > > >
> > > > > > Firstly, **GC ViT** models significantly outperform corresponding **Swin Transformer** and **ConvNext** models, hence validating its robustness on more challenging datasets with real-world images. Secondly, **GC ViT-Large** significantly outperforms **GC ViT-Base** by **+3.2%** in terms of accuracy.
> > > > > >
> > > > > > Given the performance of **GC ViT-Large** on two additional datasets of **ImageNet-A** and **ImageNet-V2**, **GC ViT-Large** scales well and its effectiveness is validated. In addition, as mentioned in the previous comment, **ImageNet-21K** pretrained **GC ViT-Large** achieves a Top-1 accuracy of **86.5%** on **ImageNet-1K**, upon further finetuning. As a results, it also scales well to **larger dataset** sizes.
> > > > > >
> > > > > > > **ImagNet-v2 results is it the max accuracy during training or the performance of the best model on ImageNet-val**
> > > > > >
> > > > > > **ImageNet-V2** results are obtained without training and by **evaluation of the best models** on **ImageNet-1K** as also reported in Table 1. The performance of the last and maximum accuracy are very similar.
> > > > > >
> > > > > > > **logs**
> > > > > >
> > > > > > The provided logs are the only ones that were available to be shared.
> > > > > >
> > > > > > We hope that our detailed responses in further discussions with the reviewer have sufficiently addressed their concerns. Evidently, the main concerns such as lack of novelty and comparison with EdgeViT have not been further brought up by the reviewer in the follow up discussions. Hence, we assume that these concerns have been resolved. In addition, as discussed in the previous comments, we have taken the reviewer's suggestion into consideration and updated the manuscript accordingly.
> > > > > >
> > > > > >
> > > > > > [1]: Wightman, R., Touvron, H. and Jégou, H., 2021. Resnet strikes back: An improved training procedure in timm. arXiv preprint arXiv:2110.00476.
> > > > > >
> > > > > > [2]: Recht, B., Roelofs, R., Schmidt, L. and Shankar, V., 2019, May. Do imagenet classifiers generalize to imagenet?. In International Conference on Machine Learning (pp. 5389-5400). PMLR.
> > > > > >
> > > > > > [3]: Hendrycks, D., Zhao, K., Basart, S., Steinhardt, J. and Song, D., 2021. Natural adversarial examples. In Proceedings of the IEEE/CVF Conference on Computer Vision and Pattern Recognition (pp. 15262-15271).

---

> > > > > > ### Author Response · Authors · 2022-12-06
> > > > > > **GC ViT-Large Performance on ImageNet-1K and ImageNet-v2**
> > > > > >
> > > > > > To verify the effectiveness of **GC ViT-Large**, we have re-trained this model on ImageNet-1K dataset and achieved a Top-1 accuracy of **84.8**. In this experiment, we **use the last checkpoint** which has a negligible 0.03% difference with the best model performance.
> > > > > >
> > > > > > In addition, to test the robustness and generalizability of the model, we have tested GC ViT-Large on ImageNet-v2 dataset and achieve an accuracy of **74.9** (Matched Frequency sampler). We summarize the results in the table below:
> > > > > >
> > > > > > | Model   | ImageNet-1K | ImageNet-v2 (MF) | ImageNet-v2 (Th-0.7) |
> > > > > > |-------------------|----------------|----------------|--------|
> > > > > > | GCViT-XXT |79.8  |69.3 | 77.2|
> > > > > > | GCViT-XTiny |82.0  | 71.3 | 78.8 |
> > > > > > | GCViT-Tiny|83.4  |73.1 | 80.5 |
> > > > > >  |GCViT-Small |83.9 |73.8  | 80.6 |
> > > > > >  |GCViT-Base |84.4 |74.4  | 81.1 |
> > > > > >  |GCViT-Large |**84.8** |**74.9**  | **81.9** |
> > > > > >
> > > > > > MF and Th-0.7 denote Matched Frequency and Threshold-0.7 samplers, respectively.
> > > > > >
> > > > > > According to the presented benchmarks, GCViT-Large outperforms GCViT-Base by **0.4%** and **0.5%** on ImageNet-1K and ImageNet-v2 datasets, respectively. Hence, these results demonstrate the **scalability** and **generalizability** of the **GCViT-Large** model. Note that these results are generated by using the last checkpoint.
> > > > > >
> > > > > > We have updated the GCViT-Large model checkpoint and training log in the repository that is exclusively provided for reviewers. In addition, we provide an external link to GCViT-Large training log for other readers in the following:
> > > > > >
> > > > > > https://tinyurl.com/5n87jb4h
> > > > > >
> > > > > > These benchmarks address the claims regarding the performance of GC ViT-Large as well as the impact of the last and best model performance. On the latter, for other model variants, we have previously shown that performance of best and last epoch are equivalent in our benchmarks.

---

> ### Author Response · Authors · 2022-12-09
> **Response to Post-rebuttal Comments**
>
> We would like to respectfully note that the reviewer has introduced new concerns in each stage of the rebuttal without engaging in further discussions to our feedback for previous comments. The score is also reduced from 3 to 1 post-rebuttal. Yet, We provide detailed responses to the reviewer’s comments post rebuttal in the following.
>
> > **Maximum vs last epoch performance.**
>
> For the benchmarks on ImageNet-1K dataset, the maximum and last epoch accuracy for all models are very similar, as the variations happen only on the 2nd decimal. Due to this reason, we only report accuracy with the **1st decimal** to mitigate these issues. For instance, the maximum and last epoch accuracy of **GC ViT-Large** are **83.83** and **83.80**. Hence, we report an accuracy of **83.8**. Similar trends are seen for other models as well.
>
> > **Performance of GC ViT-Large on ImageNet-v2**
>
> In all of our experiments, GC ViT-Large consistently outperformed GC ViT-Base model on both ImageNet-1K and ImageNet-v2. We have additionally trained another GC ViT-Large model to further investigate this point and show that GC ViT-Large outperforms GC ViT-Base on ImageNet-1K and ImageNet-v2 by **+0.4%** and **+0.5%** respectively. In addition, on **ImageNet-A** dataset, **GC ViT-Large** outperforms **GC ViT-Base** by **+3.2%**. These results validate the scalability of the **GC ViT-Large** and robustness when evaluated on other datasets.
>
>  > **Training Logs**
>
> We have provided training logs for all 6 variants of GC ViT models, in addition to model checkpoints and code repository to facilitate the reproducibility of our work. These logs are also listed in the following table:
>
> | Model   | Acc. (ImageNet-1K) | Acc. (ImageNet-v2)|Log|
> |-------------------|----------------|----------------|----------------|
> | GC ViT-XXTiny |79.8  |69.3 |https://tinyurl.com/d5dfaczc|
> | GC ViT-XTiny |82.0  | 71.3 |https://tinyurl.com/zyu6raaa|
> | GC ViT-Tiny|83.4  |73.1 |https://tinyurl.com/6wzmdcpt|
>  |GC ViT-Small |83.9 |73.8  |https://tinyurl.com/mwucr4cs|
>  |GC ViT-Base |84.4 |74.4  |https://tinyurl.com/2p9b5dca|
>  |GC ViT-Large |**84.8** |**74.9**  |https://tinyurl.com/msrzn5zj|
>
> > **Optimize performance than understand component**
>
>
> We have provided numerous ablation studies to understand the role of each component. For starters, please see **Table 4** which shows the effectiveness of each component on classification, detection and segmentation tasks. In **Table 5**, we study the role of the downsampler. In **Table S.2**, we study the effectiveness of the proposed **global query** for various tasks. In **Table S.5**, we explore the effectiveness of standalone **Global Context (GC)** module in Swin Transformer. Hence, aside from our SOTA results, we have extensively investigated the roles of each component in various scenarios.

---

### Official Review · Reviewer_YBSt · 2022-10-27

**Confidence:** 4
**Correctness:** 4
**Technical Novelty And Significance:** 2
**Empirical Novelty And Significance:** 2
**Recommendation:** 6

**Clarity, Quality, Novelty And Reproducibility:**

This paper is easy to follow. The novelty of this paper is somehow limited. I think researchers can reproduce this paper easily.

**Strength And Weaknesses:**

Strength
1. The experiments show this method is effective and can be generalized to many different tasks.
2. The problem is essential for network architecture design and the method is straightforward.
3. The visualization results show the effectiveness of this method.

Weakness
1. The combination of local and global context is a long-standing problem and has been explored by many methods. The method for merging local and global contexts lacks novelty.
2. The improvement between the proposed method and other SOTA methods is minor.

**Summary Of The Paper:**

In this paper, the authors propose a novel module to combine the local and global context. The key of this method is to introduce global query tokens into the local context module. The advantage of this method is that it can merge the context information from short and long-term ranges. The authors do experiments for image classification, detection, and segmentation tasks to show the effectiveness of the method.

**Summary Of The Review:**

Please follow the previous parts.

---

> ### Author Response · Authors · 2022-11-17
> **Response to Reviewer YBSt**
>
> We greatly appreciate the reviewer’s efforts in reviewing our work and providing valuable feedback. We appreciate the fact that the reviewer finds our method to be “effective” and “generalizable” across many different tasks. As the reviewer has correctly mentioned, our paper is an attempt to solve an essential problem in network architecture design for capturing long-range spatial dependencies. We address the reviewer’s comments in the following.
>
>
> > **The improvement between the proposed method and SOTA is minor.**
>
> We thank the reviewer for their comment. We believe that capturing global context and modeling long-range spatial dependencies has not been properly addressed in the literature, especially for transformers and ViT-based models. After the emergence of ViT as a viable computer vision backbone, it became clear that its isotropic architecture may not be suitable for tasks that require multi-scale feature representations learning. Hence, the hierarchical ViT-based models such as Swin Transformer gained traction. Yet, these models compute attention in a local manner, without the capability to learn long-range spatial information due to their limited receptive field. For instance Swin Transformers computes attention in partitioned windows and proposed window shifting for cross-region modeling, But window shifting is sub-optimal due to the limited coverage of overlapping windows.
>
> In this work, we propose a novel formulation to address these issues. Unlike previous works, we propose to only extract global query tokens, from the entire image space, and compute its interaction with local key and value tokens. In essence, this allows for capturing global context and model cross-window interactions in an efficient manner which has not been possible in previous efforts.
>
>
> > **The improvement between the proposed method and SOTA is minor.**
>
> We appreciate the reviewer’s comment. We would like to indicate that ImageNet classification is arguably one of the most contested benchmarks in computer vision, Hence, achieving new SOTA, even by small margins is difficult. However, our model still manages to substantially improve the previous benchmarks for popular SOTA models  such as ConvNeXt and Swin Transformers, sometimes by large margins. Specifcally,  **GC ViT-T**, **GC ViT-S** and **GC ViT-B** outperform comparable **Swin Transformers** variants by **+2.1%**,**+0.8%** and **+1.1%** and **ConvNeXt** variants by **+1.3%**, **+0.7%** and **+0.6%** respectively. As a result, we believe that these results should validate the effectiveness of our model, and hopefully provide more insights on the design of general computer vision backbones with SOTA performance.

---

### Author Response · Authors · 2022-12-08
**Summary of Responses**

We thank the reviewers and ACs for their time, efforts in reviewing our work. We are glad to see that reviewers find our proposed methodology to be **“effective”** and **“generalizable”** across many different tasks, **“straightforward”** with clear **“quantitative improvements”** over previous efforts, experiments to be **“detailed enough”** to verify the effectiveness of the proposed approach and the overall presentation of our work to be **“high quality”**.

We provide a summary of updates, added results and discussions per reviewers' suggestions:

1. **Code/Logs**: To facilitate the reproducibility of our work, we have anonymously and exclusively provided our code repository, all trained checkpoints and **all training logs** for reviewer's perusal.


2. **GC ViT-Large**: We have re-trained **GC ViT-Large** model on ImageNet-1K dataset and further improved the Top-1 accuracy from **84.6** to **84.8**, hence widening the performance gap compared to similar models such as ConvNext-Large with a Top-1 accuracy of  **84.3**.


3. **Scalability**: To demonstrate the scalablity of **GC ViT-Large** model, we have pre-trained and fine-tuned it on ImageNet-21K and ImageNet-1K datasets, respectively and achieved a Top-1 accuracy of **86.5**.


4. **Generalizability**: We have tested all **GC ViT** model variants on external datasets of **ImageNet-v2** [1] and **ImageNet-A** [2] to validate the generalizability and effectiveness of our proposed models. As shown below, **GC ViT** models demonstrate great robustness for all model variants on **ImageNet-v2** dataset.

| Model   | Acc. (ImageNet-1K) | Acc. (ImageNet-v2)|
|-------------------|----------------|----------------|
| GC ViT-XXTiny |79.8  |69.3 |
| GC ViT-XTiny |82.0  | 71.3 |
| GC ViT-Tiny|83.4  |73.1 |
 |GC ViT-Small |83.9 |73.8  |
 |GC ViT-Base |84.4 |74.4  |
 |GC ViT-Large |**84.8** |**74.9**  |

As shown in the following, on **ImageNet-A** dataset, GC ViT models significantly outperform Swin Transformer and ConvNeXt counterparts by a large margin, hence validating the effectiveness and robustness of our proposed model for challenging adversarial cases.

| Model   | Acc. (ImageNet-A) |
|-------------------|----------------|
| Swin-Base  |35.8 |
| ConvNeXt-Base  |36.7 |
| **GC ViT-Base**|**42.6** |
| ConvNeXt-Large  |41.1 |
| **GC ViT-Large**  |**45.8** |


5. **Plug-and-Play GC module**: Following the reviewer's suggestion, we have validated effectiveness of the Global Context (GC) module in other models. Specifically, we have used the GC module with **Swin Transformer**. As shown below, we have observed that using GC module improves the Top-1 accuracy by **+0.9%** and **+0.7%** for **Swin Transformers** **Tiny** and **Small** variants respectively. Hence, other models can benefit from the GC module as a plug-and-play component.


| Model   |Added Component |Top-1|
|-------------------|--------|--------|
| Swin-Tiny  |None| 81.3|
| Swin-Tiny| GC Module|82.2|
| Swin-Small  | None|83.0|
| Swin-Small| GC Module|83.7|

6. **SOTA Detection/Segmentation Models**: According to the reviewer's suggestion, we have added support for employing GC ViT as the backbone for DINO [3], H-Deformable-DETR [4] and Mask2Former [5] models.

> **H-DETR**

We have validated the effectiveness of our model by using **GC ViT-Tiny** in this framework and training for **36 epochs** on **MS COCO**. Our model achieves an AP of **53.4** and outperforms the counterpart with **Swin-Tiny** with an AP of 53.2.

> **DINO**

In addition, we have used a **GC ViT-Large** model, pre-trained on **ImageNet-21K dataset** as the backbone with **DINO-4 scale** and our model achieves a **box AP** of **56.4** on **MS COCO** dataset (see below).

| Model   | Epoch|Scale|AP | AP_50|AP_75|AP_S|AP_M|AP_L|
|-------------------|----------------|----------------|----------------|----------------|----------------|----------------|----------------|----------------|
| Dynamic DETR |12|5|42.9  |61.0 |46.3 |24.6 |44.9 |54.4|**0.724**|
| DINO |12|4|49.0  |66.6 |53.5 |32.0 |52.3 |63.0|**0.724**|
| **GC ViT-DINO (Ours)** |12|4|**56.4**  |**74.6** |**61.6** |**39.3** |**59.5** |**72.4**|**0.724**|

The performance of **GC ViT-DINO** in this experiment is close to SOTA on this task, and hence it validates the **scalability** and **effectiveness** of the **GC ViT-Large** model in downstream tasks.

We will release the code for all 3 models upon the acceptance of our work.

[1] Recht, B., Roelofs, R., Schmidt, L. and Shankar, V., 2019, May. Do imagenet classifiers generalize to imagenet?. In International Conference on Machine Learning (pp. 5389-5400). PMLR.

[2] Hendrycks, D., Zhao, K., Basart, S., Steinhardt, J. and Song, D., 2021. Natural adversarial examples. In Proceedings of the IEEE/CVF Conference on Computer Vision and Pattern Recognition (pp. 15262-15271).

[3] https://github.com/IDEA-Research/DINO

[4] https://github.com/HDETR/H-Deformable-DETR

[5] https://github.com/facebookresearch/Mask2Former

---

### Decision · Program_Chairs · 2023-01-20

**Decision:**

Reject

**Justification For Why Not Higher Score:**

The weakness section of the meta review describes this points in details.

**Justification For Why Not Lower Score:**

N/A

**Metareview: Summary, Strengths And Weaknesses:**

This paper introduces the GC ViT architecture that combines local and global contexts using global query tokens. The architecture includes a parameter-efficient downsampling module. The authors claim that the GC ViT architecture achieves state-of-the-art results on various tasks, including image classification, object detection, instance segmentation, and semantic segmentation.

The reviewers praised the paper for its clear exposition, simple and practical ideas, and comprehensive experimental evaluation.

The reviewers have noted that the proposed approach lacks novelty, as the community has explored similar ideas in previous work, and that the experimental results are relatively weak. The proposed architecture surpasses the comparison points, especially for model variants with small parameter counts, namely VIT-XT / VIT-T / VIT-S. Reviewers have suggested several missing and important comparison points, for instance, EdgeVit and EfficientNet. Regarding EdgeVit, the authors included results in Table 1 but did not include them in Fig. 1, claiming that EdgeVit is specialized for small model ranges. Regarding EfficientNet, the authors have refused to include the comparison. They deemed the comparison unfair, partly because of a mismatch in the number of pretraining epochs. On the other hand, GC ViT was trained for 320 epochs (see logs), while in the paper, the details describe 300, and the authors deemed the difference not having “a considerable impact on the final accuracy.” Reviewer V9pP asked for investigations about the performance of larger models, namely GC VIT-L. The performance of this model has evolved during the rebuttal phase, with the authors editing their comments along the way, making it very hard to judge and track the performance.

I have carefully read the paper, the reviews, the discussion, and all comments on OpenReview and am not making my decision based on partial information. My decision is informed by my own, independent, technical assessment of the work. In its current form, I recommend this paper for rejection. The authors should take the feedback from the reviewing process into account, improve the paper accordingly and submit it to the next venue.